# On the power and control of a misaligned rotor
# – Beyond the cosine law –

Simone Tamaro, Filippo Campagnolo, and Carlo L. Bottasso

Wind Energy Institute, Technical University of Munich, 85748 Garching b. München, Germany

**Correspondence:** Carlo L. Bottasso (carlo.bottasso@tum.de)

**Abstract.**

We present a new model to estimate the performance of a wind turbine operating in misaligned conditions. The model is based on the classic momentum and lifting-line theories, considering a misaligned rotor as a lifting wing of finite span, and accounts for the combined effects of both yaw and uptilt angles.

Improving on the classical empirical cosine law in widespread use, the new model reveals the dependency of power not only on the misalignment angle, but also on some rotor design parameters and – crucially – on the way a rotor is governed when it is yawed out of the wind. We show how the model can be readily integrated with arbitrary control laws below, above and around rated wind speed. Additionally, the model also shows that a sheared inflow is responsible for the observed lack of symmetry for positive and negative misalignment angles. Notwithstanding its simplicity and insignificant computational cost, the new proposed approach is in excellent agreement with large eddy simulations (LES) and wind tunnel experiments.

Building on the new model, we derive the optimal control strategy for maximizing power on a misaligned rotor. Additionally, we maximize the total power of a cluster of two turbines by wake steering, improving on the solution based on the cosine law.

## 1 Introduction

Wind farm control by wake steering consists of deflecting the wake away from downstream rotors to boost the total power of a plant (Meyers et al., 2022). The effectiveness of this control strategy has been proven numerically (Jiménez et al., 2010), experimentally in the wind tunnel (Campagnolo et al., 2016), as well as in field tests (Fleming et al., 2019; Doekemeijer et al., 2021). At the core of the power-boosting ability of wake steering is a trade-off: on the one hand, there is an enhanced momentum of the inflow at a downstream turbine when a wake is shifted laterally away from it; on the other hand, some power is lost at the upstream misaligned rotor, because it does not point into the wind anymore. In general the trade-off budget is positive, in the sense that the power that is gained downstream is larger than the one lost upstream. The problem is however highly complex: downstream, power capture is determined by the interaction of the impinged rotor with the wake that, in turn, is influenced by the ambient conditions and those of the wake-shedding turbine; upstream, power losses depend on the inflow characteristics, but also on the rotor and on the way it is governed. Understanding and controlling this delicate balance between upstream and downstream behavior is clearly of paramount importance for improving the power capture of wind farms by wake steering. Great progress has been made in recent years to understand, model and control wakes (see for example the

review in Meyers et al. (2022)), i.e. on the "downstream" aspect of the problem. However, the "upstream" aspect remains much less explored and understood. How much power does a yawed turbine really lose? And what inflow, rotor and rotor-control parameters influence its behavior? It is a major ambition of this paper to try and answer these questions.

The aerodynamic power $P$ of a wind turbine is customarily written as $P = \frac{1}{2}\rho A u_\infty^3 C_P$, where $\rho$ is the air density, $A$ the rotor swept area, $u_\infty$ the ambient free-stream wind speed, and $C_P$ the power coefficient. When a turbine is misaligned with respect to the wind vector by an angle $\gamma$, the rotor-orthogonal velocity component becomes $u_\infty \cos\gamma$. Accordingly, one would expect the yaw-induced power loss to be $\eta_P = P/P_0 = \cos^3\gamma$, where $P_0$ is the aerodynamic power produced for $\gamma = 0$. Unfortunately, this is only a naive interpretation of the true behavior of a misaligned rotor, and its predictions are not confirmed by experimental and numerical observations (Liew et al., 2020). To reflect this fact, a pragmatic solution has been adopted by most of the literature, where power losses due to misalignment are assumed to obey the simple law $\eta_P \approx \cos^{p_p}\gamma$, where $p_p$ is a tunable parameter.

Unsurprisingly, since such a model is not based on actual physics, a large spread of values for $p_p$ has been reported in the literature. In wind tunnel experiments with scaled models, Campagnolo et al. (2020) measured $p_p = 2.1$, Krogstad and Adaramola (2012) and Bartl et al. (2018) reported $p_p \approx 3$, whereas Medici (2005) found a value $p_p = 2$. Numerically, Fleming et al. (2015) measured $p_p = 1.88$ on the NREL 5 MW wind turbine (Jonkman et al., 2009), whereas Draper et al. (2018) obtained values between 1.3 and 2.5 for scaled wind turbine models operating in waked inflow conditions. The power production in misaligned conditions has also been measured in multiple field tests. For example, Fleming et al. (2017) reported a value of 1.41 for an Envision 4 MW turbine; Dahlberg and Montgomerie (2005) published a range of values for $p_p$ between 1.9 and 5.1 at an offshore plant. More recently, Hulsman et al. (2022) observed $2 < p_p < 2.5$ at an onshore wind farm in the north of Germany.

The large scatter characterizing the $p_p$ coefficients reported in the literature is a relevant source of uncertainty, creating a significant hindrance to the development of wind farm control strategies, and suggesting that some relevant phenomena are not captured by the $\cos^{p_p}$ law. In hindsight, this is to be expected, because this simple model fails to explicitly represent how the power coefficient $C_P$ changes when a turbine is misaligned, and somehow absorbs this effect into the tunable exponent. Some indications that there is more to this problem than a simple power cosine law have already been reported by various authors. Based on experiments and numerical simulations, Campagnolo et al. (2023), Cossu (2021a, b) and Heck et al. (2023) suggested that the power of a misaligned rotor strongly depends on its loading, in the form of the thrust coefficient $C_T$; clearly, in turn this has a strong effect on the behavior of the wake (Cossu, 2021a, b). Other variables that have been shown to play a role on power losses are related to the inflow. Recently, Draper et al. (2018) and Liew et al. (2020) have observed that power losses in misaligned conditions differ depending on whether a rotor is waked or not. Howland et al. (2020) observed a significant influence of shear and veer, while Simley et al. (2021) measured a strong dependency on inflow speed. The behavior of power losses has also been shown to depend on the direction of yaw misalignment, and not only on its magnitude as implied by the power cosine law. This asymmetric behavior of yaw misalignment has been observed by Fleming et al. (2015); Schottler et al. (2017); Fleming et al. (2018); Campagnolo et al. (2020), among others. However, an agreement on which misalignment direction yields more or less power has not been reached yet.

In this paper, we present a new analytical model for misaligned wind turbine rotors. The proposed approach combines the classic momentum and lifting-line theories, considering a misaligned rotor as a lifting wing of finite span, in close parallel to the analysis conducted for helicopter rotors in forward flight (Johnson, 1995). While existing approaches do not explain the lack of symmetry with respect to yaw direction, the present model includes the effects of wind shear, which is shown to be the culprit for the observed break of symmetry with respect to the misalignment direction. For improved accuracy, the model also includes the effects of the uptilt angle, as it contributes to the overall misalignment of the rotor with respect to the wind vector. Thanks to this feature, the proposed methodology is also readily applicable to vertical wake steering control, which could be implemented with floating wind turbines (Nanos et al., 2022) or downwind teetering rotors. The resulting model equations are integrated over the blade span and averaged over one rotor revolution, leading to a semi-analytical formulation of negligible computational cost that can be readily coupled with engineering wake models such as FLORIS (NREL, 2023b) or PyWake (Pedersen et al., 2019). However, the model governing equations could also be integrated numerically and embedded into blade element momentum (BEM) codes (Hansen, 2015), such as the AeroDyn package (NREL, 2023a) implemented in OpenFAST (NREL, 2023c).

Very recently, Heck et al. (2023) published a misaligned rotor model based on similar arguments. However, their approach does not include the effects of shear, and therefore fails to capture the asymmetric behavior of yaw direction. More importantly, their formulation uses a modified thrust coefficient $C_T'$, which is assumed to remain constant between aligned and misaligned conditions. This hypothesis is indeed verified when the turbine operates in the partial load region. Departing from this approach, the method proposed here is based on a completely general dependency of the thrust coefficient on the misalignment angle, and therefore can readily accommodate arbitrary regulation strategies in the partial, full and intermediate regulation regions – including thrust clipping and derating (Campagnolo et al., 2023). Moreover, the model of Heck et al. (2023) cannot predict power losses higher than $\cos^3\gamma$, which have however been reported in the literature. A detailed comparison of the new proposed model and the one of Heck et al. (2023) is developed in the following pages.

The proposed semi-analytical model shows that the behavior of a misaligned rotor does not follow the $\cos^{p_P}$ law, contradicting this empirical formula in widespread use. Additionally, the new model clarifies the behavior of power capture with respect to some rotor design parameters and – even more importantly – with respect to the way a rotor is governed when it is misaligned. This is an effect that has been neglected so far, but that – as already noted by Howland et al. (2020) – most probably explains the large scatter observed by various authors. Building on the unique ability of the proposed method to handle arbitrary control policies, we derive the optimal strategy for maximizing power capture when pointing a rotor away from the wind. Finally, we implement the semi-analytical model in FLORIS and we optimize the power of a cluster of two turbines. We obtain setpoints that differ from those that can be computed with the empirical $\cos^{p_P}$ law, and that lead to a slight improvement of the cluster power.

The new models exhibits an excellent match with high-fidelity LES simulations obtained with a TUM-modified version of NREL's large eddy simulator actuator line model (LES-ALM) SOWFA (Fleming et al., 2014; Wang et al., 2019, 2018). Additionally, the model is further validated with wind tunnel data from experiments conducted with the TUM G1 scaled wind turbines (Bottasso and Campagnolo, 2022; Campagnolo et al., 2020).

The paper is organized as follows: Sect. 2 presents the new formulation, and Sect. 3 explains its implementation in an engineering wake model, including the integration with arbitrary control strategies. Next, Sect. 4 considers its validation with respect to simulated and experimental data, while Sect. 5 analyzes the effects of the new model on wake steering. Finally, Sect. 6 draws conclusions and offers an outlook towards future work.

## 2 Misalignment model

### 2.1 Frames of reference

Three reference frames are necessary to completely characterize a misaligned rotor interacting with the wind, as shown in Fig. 1: a *ground-fixed* reference frame and a *nacelle-fixed* reference frame, which together describe the relative orientation of the rotor with respect to the ground, and a *wake-deflection intrinsic frame*, which describes the relative orientation of the rotor with respect to the incoming wind vector.

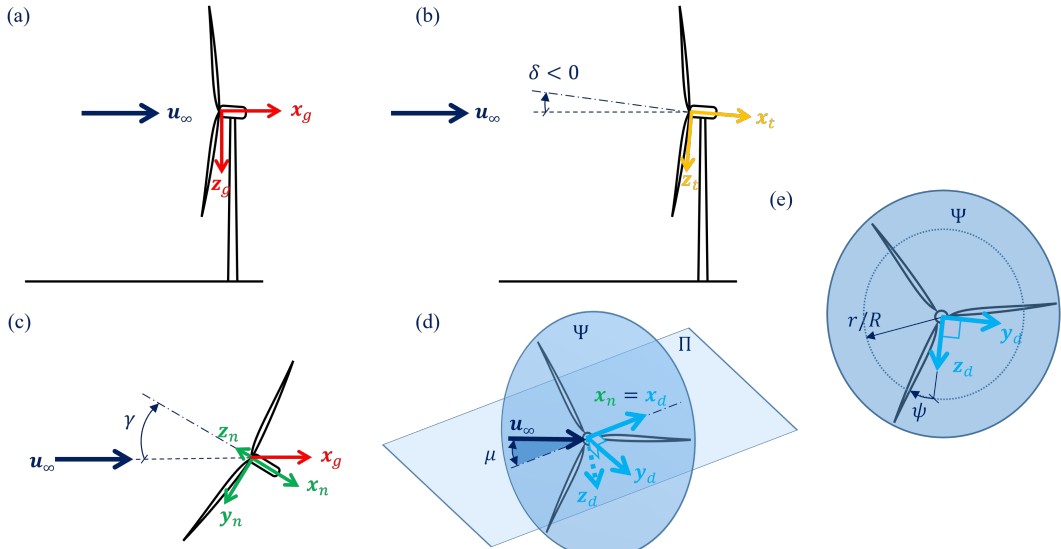

**Figure 1.** Reference frames used in the derivation of the model. Ground-fixed wind-aligned reference frame (in red, subscript $g$) **(a)**; intermediate frame, obtained by the tilt rotation $\delta$ about the horizontal axis $\boldsymbol{y}_g$ (in orange, subscript $t$) **(b)**; nacelle-fixed reference frame, obtained from $t$ by a yaw rotation $\gamma$ about the vertical axis $\boldsymbol{z}_g$ (in green, subscript $n$) **(c)**; plane $\Pi$ formed by the wind vector $\boldsymbol{u}_\infty$ and the rotor axis $\boldsymbol{x}_n$; the plane contains the $\boldsymbol{x}_d$ and $\boldsymbol{y}_d$ unit vectors of the wake-deflection intrinsic frame (in light blue, subscript $d$) **(d)**; rotor plane $\Psi$, formed by the $\boldsymbol{y}_d$ and $\boldsymbol{z}_d$ unit vectors, with rotor azimuthal angle $\psi$ and radial position $r/R$ **(e)**. Rotations are positive according to the right hand rule; notice that the value of the uptilt angle of the rotor in panel **(b)** is therefore negative.

The ground-fixed wind-aligned frame of reference is indicated with a subscript $g$ and is defined by the right-handed triad of unit vectors $\mathcal{F}_g = \{\boldsymbol{x}_g, \boldsymbol{y}_g, \boldsymbol{z}_g\}$. $\boldsymbol{z}_g$ points vertically down towards the ground, $\boldsymbol{x}_g$ is parallel to the terrain pointing downstream and is contained in the plane formed by the wind vector $\boldsymbol{u}_\infty$ and $\boldsymbol{z}_g$; finally, $\boldsymbol{y}_g$ completes a right-handed triad. In the following, for simplicity we consider the wind vector to be parallel to the terrain, i.e. $\boldsymbol{u}_\infty \parallel \boldsymbol{x}_g$, although this is not strictly necessary.

The nacelle-fixed frame of reference is indicated with a subscript $n$ and is defined by the triad of unit vectors $\mathcal{F}_n = \{\boldsymbol{x}_n, \boldsymbol{y}_n, \boldsymbol{z}_n\}$. $\mathcal{F}_n$ is obtained from $\mathcal{F}_g$ by two successive rotations: a first rotation by the *tilt* angle $\delta$ about the horizontal axis $\boldsymbol{y}_g$, followed by a second rotation by the *yaw* angle $\gamma$ about the vertical axis $\boldsymbol{z}_g$. Both rotations are positive about their respective axes according to the right hand rule (notice that, according to this definition, the typical uptilt of an upwind turbine results in a negative value for $\delta$).

However, the interaction of the rotor with the flow depends only on their mutual orientation, and not on how they are oriented with respect to the ground, which is a fundamental principle of fluid mechanics known as Galilean relativity. Therefore, a third frame is necessary, which is termed here *wake-deflection intrinsic frame* and is indicated with a subscript $d$. The frame is formed by a right-handed triad of unit vectors $\mathcal{F}_d = \{\boldsymbol{x}_d, \boldsymbol{y}_d, \boldsymbol{z}_d\}$. Vector $\boldsymbol{x}_d$ is parallel to the rotor axis, i.e. $\boldsymbol{x}_d = \boldsymbol{x}_n$, while vectors $\boldsymbol{y}_d$ and $\boldsymbol{z}_d$ are contained in the rotor disk plane $\Psi$. Together, the rotor axis $\boldsymbol{x}_d$ and the wind velocity vector $\boldsymbol{u}_\infty$ define

the $\Pi$ plane. The angle in the $\Pi$ plane between these two vectors is the *true misalignment angle* $\mu$:

$$\cos\mu = \frac{\boldsymbol{u}_\infty}{u_\infty} \cdot \boldsymbol{x}_n, \tag{1}$$

where $u_\infty = |\boldsymbol{u}_\infty|$ is the scalar ambient wind speed. The unit vector $\boldsymbol{z}_d$ is orthogonal to the $\Pi$ plane, i.e. $\boldsymbol{z}_d = \boldsymbol{x}_d \times \boldsymbol{u}_\infty / (u_\infty \sin\mu)$, while unit vector $\boldsymbol{y}_d$ is finally chosen to form a right-handed triad. Using the coordinate transformations in Appendix A, it can be readily shown that $\cos\mu = \cos\delta\cos\gamma$, i.e. the total misalignment is caused by both the tilt and yaw angles, the former

typically being neglected in most wake models. Notice that, given its definition, the misalignment angle $\mu$ is always positive, because $\boldsymbol{z}_d$ flips from one side of the $\Pi$ plane to the other, depending on the relative orientation of the wind velocity and rotor axis vectors. When the wind comes from the right looking upstream in the $\Pi$ plane, $\boldsymbol{z}_d$ points downwards (see Fig. 1d), whereas it points upwards when the wind comes from the left.

Figure 2 shows a visualization of the wakes developing behind a wind turbine rotor for two different pairs of tilt and yaw

values: $\delta = 0°$, $\gamma = -30°$; and $\delta = -28.43°$, $\gamma - 10°$. Both pairs correspond to a same true misalignment $\mu = 30°$. The figure confirms that the wake is invariant for an observer on the $\mathcal{F}_d$ frame. This is particularly evident in the images of the longitudinal speed on the $\Pi$ plane (marked with a black solid border), which are clearly identical in the two cases. Clearly, for large values of tilt the interaction of the wake with the ground or with a sheared inflow would break the $\Pi$-frame invariance.

Because of what noted above, in the following the wake analysis is developed in the $\Pi$ plane, instead of the horizontal

one as customarily done. Transformation matrices that map vector components from one frame to the other are reported in Appendix A.

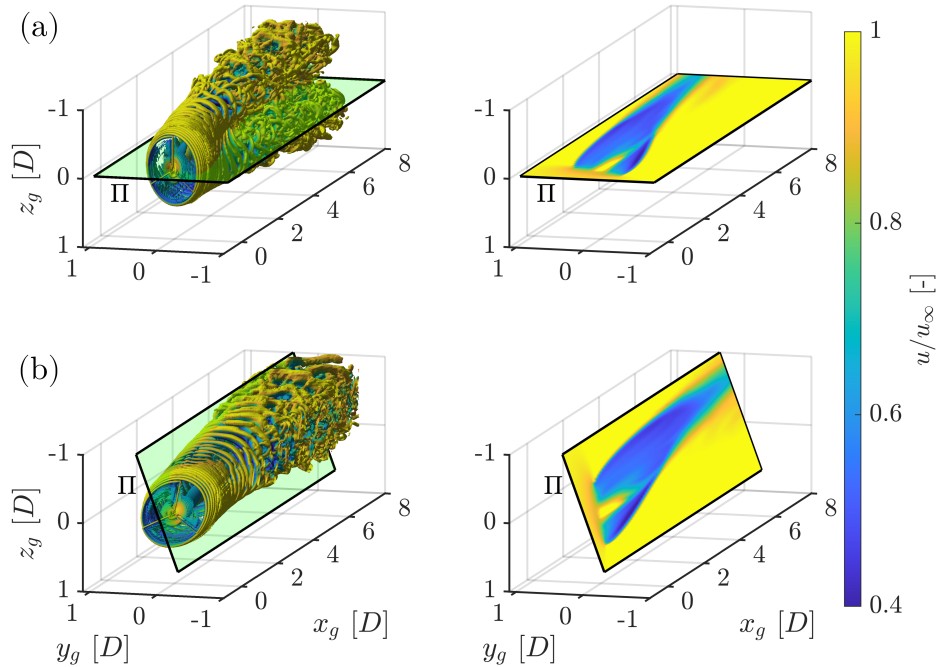

**Figure 2.** Visualization of the wakes developing behind a wind turbine operating in steady inflow conditions for different pairs of tilt and yaw values, all corresponding to a same total (true) misalignment $\mu = 30°$. The deflection of the wake occurs in the $\Pi$ plane, marked with a black border. Top row **(a)**: $\delta = 0°$, $\gamma = -30°$; bottom row **(b)**: $\delta = -28.43°$, $\gamma = -10°$. Left column: iso surfaces of $Q$-criterion; right column: image of the longitudinal flow speed $u/u_\infty$ on the $\Pi$ plane. Distances are expressed in rotor diameters $D$. Interactive 3D versions of the figures are available at the following links: https://tinyurl.com/btcl-fig-2-a **(a)**; https://tinyurl.com/btcl-fig-2-b **(b)**.

## 2.2 Sheared inflow

Considering a linear vertical shear of the inflow, the ambient wind speed writes

$$u_\infty(z_g) = u_{\infty,\text{hub}} \left(1 - k\frac{z_g}{R}\right). \tag{2}$$

Here $u_{\infty,\text{hub}}$ is the ambient wind speed at hub height, $k$ is the vertical linear shear coefficient, $z_g$ is the vertical coordinate in the ground frame of reference, centered at the hub, and $R$ is the rotor radius. The choice of a linear shear distribution was made just to simplify the derivations, and other choices are clearly possible, for example to model the more common power law or the presence of low-level jets. Additionally, it would be interesting to include also the effects of a horizontal shear, to account for waked conditions, and of veer. These further model improvements are however deferred to a continuation of this study.

By applying the coordinate transformation of Appendix A, the ambient wind speed of Eq. (2) can be written in terms of the radial $r$ and azimuthal $\psi$ coordinates on the rotor plane, yielding

$$u_\infty(r, \psi, \delta, \gamma) = u_{\infty,\text{hub}} \left(1 - k\frac{r}{R}\frac{\cos\delta}{\sin\mu}\left(\sin\gamma\cos\psi - \cos\gamma\sin\delta\sin\psi\right)\right). \tag{3}$$

Here $\psi$ is positive about $x_d$ according to the right hand rule (i.e. clockwise looking downstream), and it is measured starting from the $z_d$ unit vector (which flips from one side of the $\Pi$ plane to other depending on whether the wind blows from the right or left looking upstream, as explained in Sect. 2.1; see also Fig. 1).

## 2.3 Force and velocity components at a blade section

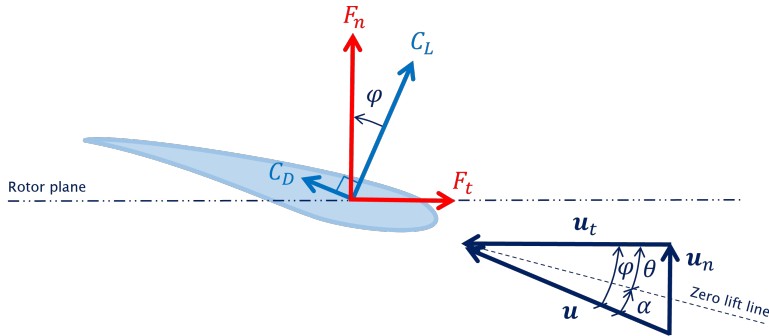

**Figure 3.** Blade cross section, with triangle of velocities (in blue), lift and drag (in light blue), and resulting aerodynamic force components (in red).

With reference to Fig. 3, the tangential $F_t$ and normal $F_n$ components of the aerodynamic force at a blade section are

$$F_t = \frac{1}{2}\rho u^2 c \left(C_L \sin\varphi - C_D \cos\varphi\right), \tag{4a}$$

$$F_n = \frac{1}{2}\rho u^2 c \left(C_L \cos\varphi + C_D \sin\varphi\right), \tag{4b}$$

where $\varphi = \tan^{-1} u_n/u_t$ is the inflow angle, $u = \sqrt{u_t^2 + u_n^2}$ is the total flow speed at the blade section, $c$ is the sectional chord length, and finally $C_L$ and $C_D$ are the lift and drag coefficients, respectively. Using the coordinate transformations of Appendix A, the tangential $u_t$ and normal $u_n$ velocity components write

$$u_t = \Omega r + u_\infty \sin\mu \cos\psi, \tag{5a}$$

$$u_n = u_\infty \cos\mu (1 - a), \tag{5b}$$

where $\Omega$ is the angular speed of the rotor and $a$ is the axial induction factor, which expresses how much the rotor-orthogonal component of the free-stream speed $u_\infty$ is slowed down at the rotor disk.

## 2.4 Induction model

It is well known that a non-uniform description of the induction is necessary in order to accurately capture the azimuthal variation of loads on a rotor operating in non-axial conditions (Johnson, 1995). However, it appears that this is not necessary

when computing integral rotor quantities such as power, torque and thrust, as in the present case. To show this, the induction is modeled here with an expansion limited to one-per-revolution (1P) harmonics, i.e.

$$a = a_0 \left( 1 + \kappa_{1s} \frac{r}{R} \sin \psi + \kappa_{1c} \frac{r}{R} \cos \psi_g \right), \tag{6}$$

where $a_0$ is the constant-over-the-rotor (0P) induction, while $a_0 \kappa_{1s}$ and $a_0 \kappa_{1c}$ are the 1P sine and cosine harmonic amplitudes, respectively.

Following the classical approach used for helicopter rotors in forward flight (Johnson, 1995), the sine term accounts for the tilting of the induction plane caused by the misalignment $\mu$ of the rotor with the incoming wind. As such, it is written in terms of $\psi$, which is measured starting from the $z_d$ unit vector, and therefore it expresses a rotation of the induction plane about the axis normal to the wake-deflection intrinsic frame $\Pi$. The coefficient $\kappa_{1s}$ can be modeled according to Coleman et al. (1945) and Pitt and Peters (1981), resulting in the expression

$$\kappa_{1s} = -\frac{15\pi}{32} \tan \left( \frac{\chi}{2} \right), \tag{7}$$

where the initial wake skew angle is $\chi = \mu + \sin \mu \, C_T / 2$ (Jiménez et al., 2010), and $C_T = 2T/(\rho A u_{\infty,\text{hub}}^2)$ is the thrust coefficient. Notice that the definition of the skew angle differs from the one given by Eq. (20) of Jiménez et al. (2010), because of the different definition of the thrust coefficient used in that publication.

The cosine term is introduced to account for the effects on the induction caused by vertical shear. As such, it is written as a function of the azimuthal angle $\psi_g$, which is measured from the (vertical) $z_g$ unit vector, and therefore it expresses a rotation of the induction plane about the (horizontal) unit vector $y_g$. Using Eq. (A4b), it is readily found that $\psi_g = \psi \cos \mu$. Following Meyer Forsting et al. (2018), the cosine term is proportional to both the shear $k$ and the thrust $C_T$ coefficients, i.e.

$$\kappa_{1c} = \kappa_{1c}^* k \, C_T. \tag{8}$$

The cosine term significantly complicates the analytical derivations of power, torque and thrust, which must now be expressed in terms of Bessel functions (Abramowitz et al., 1988) because of the term $\cos(\psi \cos \mu)$. Before attempting the modeling of the proportionality coefficient $\kappa_{1c}^*$, this term was numerically optimized to best fit the numerical simulations and experimental measurements, as explained later in Sect. 3.

The inclusion of the sine and cosine induction terms has only an extremely modest effect on the quality of the results. In fact the match of $C_P$ improves by 0.35% when the sine term is included, and by 0.60% when both terms are used, as more precisely shown in Appendix B. Because of their modest effects, these terms are dropped from the following discussion, to simplify the resulting expressions, and they were not used in the results reported later in this article. However, these terms are retained in the software implementation of the model (Tamaro et al., 2024), and can be switched on if desired by the user.

It should also be noted that the present model neglects the effects of the tangential induction, which in fact does not appear in the tangential velocity component expressed by Eq. (5a). This is justified by the fact that the rotor swirl is concentrated close to the hub, and it is small for a large extent of the blade span (Burton et al., 2011), where most of the thrust and power are generated.

More in general, there are several other effects that are present in a rotor, and that are not modelled here, such as for example radial drag, tip and root losses, blade sweep, prebend and cone, and others. All these effects can be taken into account in detailed BEM models (Hansen, 2015; Burton et al., 2011), but would significantly complicate the present simplified analytical method. However, notwithstanding these limitations, the results of Sect. 4 show a remarkable ability of the proposed approach in predicting the trends of power and thrust as functions of various operating and inflow conditions. Additionally, the ability of the model in predicting actual power and thrust values (instead of trends) can be improved by the use of loss functions, as explained in Sect. 3.2.

## 2.5 Streamtube model

An expression for the axial induction can be derived using the concept of a streamtube (Hansen, 2015), as shown in Fig. 4 with reference to the present case of a misaligned rotor. Four stations are considered along the stream tube: inlet $i$; outlet $o$; section $r^-$ located immediately in front of the rotor; and section $r^+$ located immediately behind the rotor.

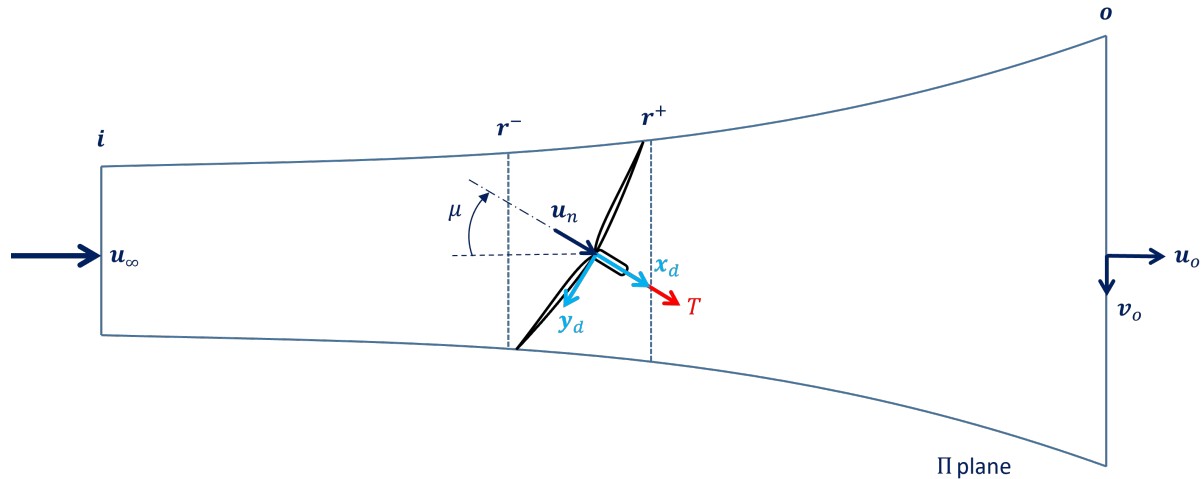

**Figure 4.** Schematic view of a streamtube around a misaligned wind turbine. Cross-sectional stations: inlet $i$, outlet $o$, $r^-$ immediately in front of the rotor; $r^+$ immediately behind the rotor.

The principle of impulse and momentum applied to the streamtube is written as

$$T \cos \mu = \dot{m}(u_\infty - u_o), \tag{9}$$

where $T$ is the thrust force, $u_o$ is the longitudinal flow speed at the streamtube outlet, while $\dot{m}$ is the mass flux

$$\dot{m} = \rho A u_n. \tag{10}$$

By using the thrust coefficient, Eq. (9) yields the non-dimensional longitudinal flow speed at the streamtube outlet:

$$\frac{u_o}{u_{\infty,\text{hub}}} = 1 - \frac{1}{2}\frac{C_T}{1-a_0}. \tag{11}$$

Next, Bernoulli's energy conservation theorem is applied between the streamtube inlet and the section immediately upstream of the rotor (stations $i$ and $r^-$ in Fig. 4), and between the section immediately downstream of the rotor and the streamtube outlet (stations $r^+$ and $o$ in the same figure):

$$p_i + \frac{1}{2}\rho V_i^2 = p_{r^-} + \frac{1}{2}\rho V_{r^-}^2, \tag{12a}$$

$$p_{r+} + \frac{1}{2}\rho V_{r+}^2 = p_o + \frac{1}{2}\rho V_o^2, \tag{12b}$$

where $p$ is pressure and $p_i = p_o = p_\infty$, while $p_\infty$ is the ambient value. Additionally, $V_{r^-} = V_{r^+}$ for continuity, and furthermore $V_o^2 = u_o^2 + v_o^2$ at the outlet section, where $v_o$ is the lateral (sidewash) speed component.

Following a customarily text-book assumption used for helicopters rotors in forward flight (Johnson, 1995), only more recently adopted also for wind turbines by Shapiro et al. (2018), the misaligned rotor can be seen as a lifting wing of finite span (albeit of a small aspect ratio $AR = D^2/A = 4/\pi$) operating at an angle of attack $\mu$[1]. The chord $C(y_d)$ of the wing in the streamwise direction has an elliptic distribution: $(C(y_d)/2)^2 + y_d^2 = R^2$. According to Prandtl's lifting line theory (Tietjens and Prandtl, 1957; Katz and Plotkin, 2001), the wing has consequently an elliptic lift distribution, which induces a spanwise-constant downwash (in this case, sidewash) $v_o = \Gamma/4R$. $\Gamma = \bar{L}/\rho u_{\infty,\text{hub}}$ is the circulation at the wing mid section $y_d = 0$, and $\bar{L} = LC(0)/A$ is the lift per unit span at that same location. Since the wing lift is the rotor side force, i.e. $L = T\sin\mu$, it follows that the non-dimensional sidewash at the streamtube outlet can be expressed as (Heck et al., 2023)

$$\frac{v_o}{u_{\infty,\text{hub}}} = \frac{1}{4}C_T\sin\mu. \tag{13}$$

Combining the previous equations, yields an expression for the 0P axial induction $a_0$ as a function of the misalignment $\mu$ and thrust coefficient $C_T$:

$$1 - a_0 = \frac{1 + \sqrt{1 - C_T - \frac{1}{16}C_T^2\sin^2\mu}}{2\left(1 + \frac{1}{16}C_T\sin^2\mu\right)}. \tag{14}$$

## 2.6 Thrust force

Equation (14) furnishes an expression for the 0P axial induction as a function of the thrust coefficient. To close the problem, an expression for the thrust coefficient in terms of the operating conditions of the turbine is necessary. To this end, the thrust force $T$ is expressed in terms of the normal sectional force $F_n$ as

$$T = \frac{B}{2\pi}\int_0^{2\pi}\int_0^R F_n\,\mathrm{d}\psi\mathrm{d}r, \tag{15}$$

---

[1]This interpretation also reveals that the so-called curled shape of laterally deflected wakes (see e.g. Martínez-Tossas et al. (2021) and references therein) is nothing else than the effect of the horseshoe vortex structure generated behind a lifting wing, albeit with the addition of the swirl caused by the rotor rotation.

where $B$ indicates the number of blades. Using Eq. (4b) under the assumption of a small inflow angle (i.e. $\sin\varphi \approx \varphi$ and $\cos\varphi \approx 1$), yields

$$T = \frac{B}{2\pi} \int_0^{2\pi} \int_0^R \frac{1}{2}\rho u^2 c (C_D\varphi + C_L)\, \mathrm{d}\psi \mathrm{d}r. \tag{16}$$

The lift coefficient is written as $C_L = C_{L,\alpha}\alpha = C_{L,\alpha}(\alpha_g - \alpha_0)$, where $C_{L,\alpha}$ is the lift slope, and $\alpha = \varphi - \theta$ is the angle of attack measured with respect to the zero-lift direction, whereas $\alpha_g$ is the angle of attack measured with respect to a generic direction. Without any loss of generality, the use of $\alpha$ is preferred in the following, to avoid carrying along in the derivations the unnecessary extra term $\alpha_0$. Furthermore, $\theta = \theta_p + \beta$ is the local pitch angle (see Fig. 3), where $\theta_p$ is the blade pitch rotation at the pitch bearing, and $\beta$ indicates the blade twist referred to the zero-lift direction.

Neglecting swirl induction, the inflow angle is $\tan\varphi = (1 - a_0)/(\lambda r/R)$, where $\lambda = \Omega R/u_{\infty,\mathrm{hub}}$ is the tip speed ratio. For a turbine operating close to optimal induction (i.e., $a_0 = 1/3$) and a typical tip speed ratio of 8.5, the inflow angle at three-quarter span is less than 6°, justifying the small angle assumption. This assumption clearly becomes less accurate for small inductions and tip speed ratios, or close to the blade root, where however only a modest contribution to the thrust is generated.

Using again the small inflow angle assumption, it follows that $\varphi \approx u_n/u_t$ and $u \approx u_t$, and the thrust $T$ becomes

$$T = \frac{B}{2\pi} \int_0^{2\pi} \int_0^R \frac{1}{2}\rho u_t^2 c \left( C_D \frac{u_n}{u_t} + C_{L,\alpha}\left( \frac{u_n}{u_t} - \theta \right) \right) \mathrm{d}\psi \mathrm{d}r. \tag{17}$$

Using Eqs. (3) and (5), solving the double integral and expressing $T$ through the thrust coefficient $C_T = C_{T_1} + C_{T_2}$, finally gives

$$C_{T_1} = \frac{\sigma}{2}(C_D + C_{L,\alpha})\cos\mu\,(\lambda - k\cos\delta\sin\gamma)\,(1 - a_0), \tag{18a}$$

$$C_{T_2} = -\frac{\sigma}{2}C_{L,\alpha}\theta\left( \sin^2\mu + \frac{2}{3}\lambda^2 - \frac{k\cos\delta}{12}\left( 8\lambda\sin\gamma - k\cos\delta(\cos^2\gamma\sin^2\delta + 3\sin^2\gamma) \right) \right), \tag{18b}$$

where $\sigma = BcR/A$ is the rotor solidity.

For null shear, i.e. $k = 0$, this expression simplifies to

$$C_T = \frac{\sigma}{2}\left( (C_D + C_{L,\alpha})\cos\mu\left( (1 - a_0)\lambda \right) - C_{L,\alpha}\theta\left( \sin^2\mu + \frac{2}{3}\lambda^2 \right) \right). \tag{19}$$

Notice that the terms in Eqs. (18) depending on shear $k$ also depend on the angles $\gamma$ and $\delta$, whereas Eq. (19) only depends on the total misalignment angle $\mu$. This is because the wind shear is defined with respect to the ground frame, which is mapped into the nacelle frame by the $\gamma$ and $\delta$ angles, whereas $\mu$ only depends on the relative orientation of the wind vector with the rotor axis, as explained in Sect. 2.1.

Furthermore, we note that – differently from the approach of Heck et al. (2023) – these expressions for thrust are applicable to any desired control policy, as they depend explicitly on the tip speed ratio $\lambda$ and the pitch setting $\theta$.

## 2.7 Power

The aerodynamic power $P$ generated by a wind turbine is $P = Q\Omega$, where the aerodynamic torque $Q$ writes

$$Q = \frac{B}{2\pi} \int_0^{2\pi} \int_0^R F_t \, r \, \mathrm{d}\psi \, \mathrm{d}r. \tag{20}$$

Considering small angles, power can be written as

$$P = \frac{B}{2\pi} \int_0^{2\pi} \int_0^R \frac{1}{2} \rho u_t^2 c \left( -C_D + C_{L,\alpha} \left( \frac{u_n}{u_t} - \theta \right) \frac{u_n}{u_t} \right) \Omega r \, \mathrm{d}\psi \mathrm{d}r. \tag{21}$$

Using Eqs. (3) and (5), expressing the angular velocity as $\Omega = \lambda u_{\infty,\text{hub}} / R$, and solving the double integral yields the power coefficient $C_P$:

$$C_P = \frac{\sigma}{2} \lambda \left( C_{L,\alpha}(1 - a_0) \cos\mu \left( (1 - a_0) \cos\mu - \frac{2}{3}\lambda\theta \right) - \frac{1}{2} C_D (\lambda^2 + \sin^2\mu) \right.$$

$$+ k \cos\delta \sin\gamma \left( \frac{2}{3} \cos\mu \, C_{L,\alpha}\theta(1 - a_0) - \frac{1}{2}\lambda C_D \right)$$

$$\left. + \frac{1}{4} k^2 \cos^2\delta \left( \cos^2\mu \, C_{L,\alpha}(1 - a_0)^2 - \frac{1}{4} C_D (\sin^2\mu + 2\sin^2\gamma) \right) \right). \tag{22}$$

For null shear, i.e. $k = 0$, the power coefficient simplifies to

$$C_P = \frac{\sigma}{2} \lambda \left( C_{L,\alpha}(1 - a_0) \cos\mu \left( (1 - a_0) \cos\mu - \frac{2}{3}\lambda\theta \right) - \frac{1}{2} C_D (\lambda^2 + \sin^2\mu) \right). \tag{23}$$

Due to the explicit dependency of $C_P$ on $\cos\mu$ and of $1 - a_0$ on $\sin^2\mu$ (see Eq. 14), it follows that – in an unsheared inflow – the aerodynamic power does not depend on the misalignment direction.

## 2.8 Dependency on misalignment direction

The power model reveals that vertical shear is the culprit for the observed lack of symmetry with respect to yaw misalignment. In fact, the term responsible for the asymmetry in Eq. (22) is

$$C_P^{\text{asymm}} = k \cos\delta \sin\gamma \left( \frac{2}{3} \cos\mu \, C_{L,\alpha}\theta(1 - a_0) - \frac{1}{2}\lambda C_D \right). \tag{24}$$

This term shows that yawing a rotor out of the wind in a sheared inflow will produce a non-symmetric behavior with respect to positive and negative yaw angles $\gamma$, i.e. $P(-\gamma) \neq P(+\gamma)$. In fact, the following can be noted:

- For small $\theta$, the two terms within the parenthesis of Eq. (24) are small. Consequently, the asymmetry is small. Furthermore, either positive or negative yaw angles could produce more power, depending on the balance of these two terms.

– On the other hand, for larger $\theta$ the thrust coefficient $C_T$ decreases, in turn increasing the term $(1-a_0)$. As a consequence, the first of the two terms of Eq. (24) prevails. The origin of this prevailing term can be traced to the contribution to power of the fraction of lift that depends on the pitch angle, which is proportional to $C_{L,\alpha}\theta u_t u_n$ (see Eq. 21). When the rotor is misaligned, $u_t$ exhibits a $\cos\psi$ variation (see Eq. 5a) that is in phase or in opposition with a similar $\cos\psi$ variation of $u_n$ caused by shear (see Eq. 5b and 3). The integral of the product of these two terms over one revolution is different form zero, and it has the sign of the misalignment, resulting in some extra power for positive yaw angles and some losses for negative ones, when the thrust is low.

This complex balance of effects is probably the cause for the lack of agreement in the literature on which misalignment direction yields more. As shown by the model, there is no simple answer, and the behavior depends on the rotor design and on how it is operated.

Very similar conclusions apply also to the thrust coefficient. According to Eqs. (18), $C_{T_1}(-\gamma) > C_{T_1}(+\gamma)$, because $C_{T_1}$ depends on $-k\sin\gamma$; whereas $C_{T_2}(-\gamma) < C_{T_2}(+\gamma)$ (when $\theta > 0$), because $C_{T_2}$ depends on $+\theta k\sin\gamma$. Therefore one can expect a slightly higher thrust for negative yaw angles at low pitch settings, and viceversa at the higher pitch values (the effect being more pronounced at larger tip speed ratios).

To illustrate these findings in an exemplary case, Fig. 5 shows the thrust and power coefficients as functions of the misalignment angle $\gamma$, for different shear coefficients $k$ and blade pitch angles $\theta_p$. All coefficients have been normalized by their respective value in aligned conditions.

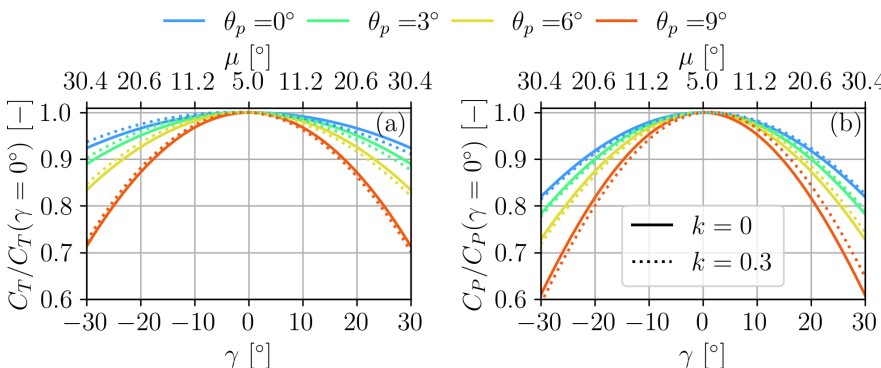

**Figure 5.** Normalized thrust $C_T/C_T(\gamma = 0°)$ **(a)** and power $C_P/C_P(\gamma = 0°)$ **(b)** coefficients, plotted as functions of the misalignment angle $\gamma$, for different shear coefficients $k$ and pitch angles $\theta_p$. Pitch values: colors; $k = 0$: solid lines; $k = 0.3$: dotted lines. The plots were generated considering the following values: $\lambda = 8.5$, $C_D = 5.2 \times 10^{-3}$, $C_{L,\alpha} = 4.76$, $\beta = 3.35°$, $\sigma = 4.16\%$, $\delta = -5°$, $R = 65$ m. An interactive version of the figure that allows one to plot the thrust and power coefficients for user-defined values of the model parameters is available as a Jupyter notebook at the link https://tinyurl.com/btcl-fig-5.

The lack of symmetry of the rotor with respect to misalignment direction is in general rather small. In a typical field imple-
mentation of wake steering, various uncertainties – e.g., due to limits in the knowledge of the ambient conditions, actual yaw
orientation of the rotor, asymmetric behavior of the onboard anemometry, etc. – and other model errors probably dominate the
problem, making the asymmetric behavior of misalignment a negligible effect, especially for small pitch values and moderate
tip speed ratios.

## 3 Implementation

### 3.1 Integration with a wake model

The analytical model derived in the previous pages can be readily implemented in engineering flow models (NREL, 2023b;
Pedersen et al., 2019). An open-source implementation in FLORIS (NREL, 2023b) is available on Github (Tamaro et al., 2024).
As a summary, we report here for convenience the governing Eqs. (14) and (18), which write

$$1 - a_0 = \frac{1 + \sqrt{1 - C_T - \frac{1}{16}C_T^2 \sin^2 \mu}}{2\left(1 + \frac{1}{16}C_T \sin^2 \mu\right)}, \tag{25a}$$

$$C_T = \frac{\sigma}{2}\Bigg( (C_D + C_{L,\alpha}) \cos\mu \left(\lambda - \cos\delta \sin\gamma \, k\right) (1 - a_0) -$$

$$C_{L,\alpha}\, \theta \left( \sin^2 \mu + \frac{2}{3}\lambda^2 - \frac{k\cos\delta}{12}\left(8\lambda \sin\gamma - k\cos\delta(\cos^2\gamma \sin^2\delta + 3\sin^2\gamma))\right) \right) \Bigg). \tag{25b}$$

This represents a closed system of equations that can be solved for the axial induction $a_0$ and thrust coefficient $C_T$, given the
tip speed ratio $\lambda$, the pitch setting $\theta_p = \theta - \beta$, and the yaw misalignment $\gamma$. Crucially, the presence of $\lambda$ and $\theta_p$ enable using
any desired control policy when misaligning the turbine. Having obtained $a_0$ and $C_T$, the power coefficient $C_P(\gamma, \theta_p, \lambda)$ is
obtained by Eq. (22) and, finally, the rotor power is computed as

$$P(\lambda, \theta_p, \gamma) = \frac{1}{2}\rho A u_{\infty,\text{hub}}^3 C_P(\gamma, \theta_p, \lambda). \tag{26}$$

### 3.2 Improved accuracy by the use of loss functions

The analytical derivation of the equations implies that the model lacks many of the features that are present in more sophis-
ticated BEM implementations, such as radially and azimuthally non-uniform induction, swirl induction, tip and root losses,
radial flow, spanwise varying geometric characteristics, prebend, etc. Clearly, this lack of accuracy could be resolved by nu-
merically implementing the same model in a BEM code (Hansen, 2015). However, this way the use of the misaligned rotor
model in combination with an engineering wake model would become much more complex and numerically expensive.

To address this problem, the following implementation is recommended, which was used in the validation and the examples reported in the following section. A power loss function $\eta_P$ is computed by using Eq. (22) to yield

$$\eta_P(\lambda,\theta_p,\gamma) = \frac{C_P(\lambda,\theta_p,\gamma)}{C_P(\lambda,\theta_p,0)}. \tag{27}$$

Next, a refined power coefficient $C_P^{\mathrm{rf}}$ is obtained as

$$C_P^{\mathrm{rf}}(\lambda,\theta_p,\gamma) = \eta_P(\lambda,\theta_p,\gamma)C_P^{\mathrm{hf}}(\lambda,\theta_p,0), \tag{28}$$

where $C_P^{\mathrm{hf}}$ is the power coefficient computed in aligned conditions through a higher-fidelity model, for example based on a sophisticated BEM implementation or even on experimental measurements, when available. In other words, the analytical model is used not to predict the actual power coefficient, but only the fraction of it that is lost by misalignment. The actual power coefficient is obtained by applying the loss model to a more accurate power coefficient model in aligned conditions.

The same approach is adopted for thrust. First, a thrust coefficient loss factor is computed by using the proposed model as $\eta_T(\lambda,\theta_p,\gamma) = C_T(\lambda,\theta_p,\gamma)/C_T(\lambda,\theta_p,0)$. Next, a refined estimate in misaligned conditions is obtained by computing $C_T^{\mathrm{rf}}$ from a higher-fidelity aligned value $C_T^{\mathrm{hf}}$, i.e.

$$C_T^{\mathrm{rf}}(\lambda,\theta_p,\gamma) = \eta_T(\lambda,\theta_p,\gamma)C_T^{\mathrm{hf}}(\lambda,\theta_p,0). \tag{29}$$

In the following, we always adopt this approach for the power and thrust coefficients. However, to simplify the notation, we drop the superscript $(\cdot)^{\mathrm{rf}}$. Hence, for example, when we write $C_P(\lambda,\theta_p,\gamma)$, we in reality imply that Eq. (28) is used; the same holds for $C_T$.

### 3.3 Implementation of arbitrary control strategies

When a wind turbine yaws out of the wind, the inflow seen by the rotor changes with respect to the aligned condition. The controller reacts to the changed inflow, modifying the setpoint, which in turn affects the power captured by the rotor and its loading. Hence, the problem is implicit, in the sense that the misalignment model has to be solved together with the controller. This general implicit approach should be contrasted with the explicit one proposed in Heck et al. (2023), which assumes a rotor performance parameter, $C_T' = 2T/(\rho A u_n^2)$, to remain constant even in misaligned conditions. This section explains how arbitrary control laws can be integrated with the present more general model.

The control of a modern variable-speed wind turbine is typically based on the definition of two or three main operational regions.

In region II (also called the below-rated or partial-load regime), the turbine should maximize its power output. This is achieved by operating at the maximum power coefficient $C_P^*(\lambda^*,\theta_p^*)$, which corresponds to the optimal tip speed ratio $\lambda^*$ and pitch setting $\theta_p^*$. As the tip speed ratio must remain constant at its value $\lambda^*$ throughout this control region, the rotor speed increases linearly with wind speed, i.e. $\Omega = \lambda^* u_\infty/R$. The aerodynamic torque $Q_a$ is readily computed as $Q_a = K(\rho)\Omega^2$, with $K(\rho) = \frac{1}{2}\rho A R^3 C_P^*/\lambda^{*^3}$. Once the aerodynamic torque is known, the torque provided by the generator $Q_g$ is obtained from the expression $Q_a = \eta_m\eta_e Q_g$, where $\eta_m$ and $\eta_e$ are the mechanical and electrical efficiencies, respectively.

When the ambient wind speed is above the rated value $u_{\infty_r} = \Omega_r R/\lambda^*$, there is enough power carried by the wind for the turbine to produce its maximum (rated) output $P_r$. This is called region III (also termed the above-rated or full-load regime), and the turbine operates at the constant (rated) rotor speed $\Omega_r$. Hence, the aerodynamic torque is constant, i.e. $Q_a = P_r/\Omega_r$, whereas blades are progressively pitched into the wind to reduce $C_P$ as wind speed increases.

To implement these standard region II and III control strategies in the proposed model, the following power equation is introduced:

$$\frac{1}{2}\rho A u_\infty^3 C_P(\lambda(\Omega), \theta_p, \gamma) = Q_a(\Omega)\,\Omega. \tag{30}$$

The equation has three unknowns: the rotor speed $\Omega$, the blade pitch $\theta_p$, and the aerodynamic torque $Q_a$. Given ambient conditions $u_\infty$ and $\rho$, two additional conditions are necessary before the three unknowns can be computed. To this end, one can first assume that the machine operates in region II. Hence, Eq. (30) is solved by appending to it the following two constraints:

$$\theta_p = \theta_p^*, \tag{31a}$$

$$Q_a = K(\rho)\Omega^2. \tag{31b}$$

If the computed rotor speed exceeds the rated value, i.e. $\Omega > \Omega_r$, then it means that – for the given ambient conditions and misalignment angle – the turbine operates in region III and not region II. Hence, the solution is discarded, and Eq. (30) is solved again by appending this time the following two constraints:

$$\Omega = \Omega_r, \tag{32a}$$

$$Q_a = P_r/\Omega_r. \tag{32b}$$

This same approach can be used for curtailment and derating strategies (Juangarcia et al., 2018).

Figure 6 shows the application of this approach to the IEA 3.4 MW reference wind turbine, a typical onshore machine with contemporary design characteristics (Bortolotti et al., 2019). In this example the turbine is exposed to an inflow characterized by a wind speed $u_\infty = 10.5$ ms$^{-1}$, an air density $\rho = 1.22$ kgm$^{-3}$, and a linear vertical shear coefficient $k = 0.2$. For these ambient conditions, the turbine operates in region III when it is aligned with the wind ($\gamma = 0°$). As the turbine starts yawing out of the wind, it initially keeps operating in region III. Accordingly, the tip speed ratio $\lambda$ (Fig. 6d) and the power coefficient $C_P$ (Fig. 6b) remain constant, while the thrust coefficient increases (Fig. 6a) and the blades pitch back (Fig. 6c). However, at around $|\gamma| \approx 15°$, the turbine enters into region II, because the rotor-orthogonal component of the wind speed is not anymore large enough to maintain the rated power output. As the misalignment keeps increasing, the pitch angle remains fixed at its optimal value (Fig. 6c), whereas the tip speed ratio drops on account of the slowing rotor speed (Fig. 6d), in accordance with the region II policy.

Often turbines present an additional intermediate operating regime, called region II1/2, which occupies a wind speed interval across the rated value $u_\infty^{lb} \leq u_{\infty_r} \leq u_\infty^{ub}$. In such cases, the turbine operates in region II when $u_\infty < u_\infty^{lb}$, in region III when $u_\infty > u_\infty^{ub}$, and in region II1/2 when $u_\infty^{lb} \leq u_\infty \leq u_\infty^{ub}$.

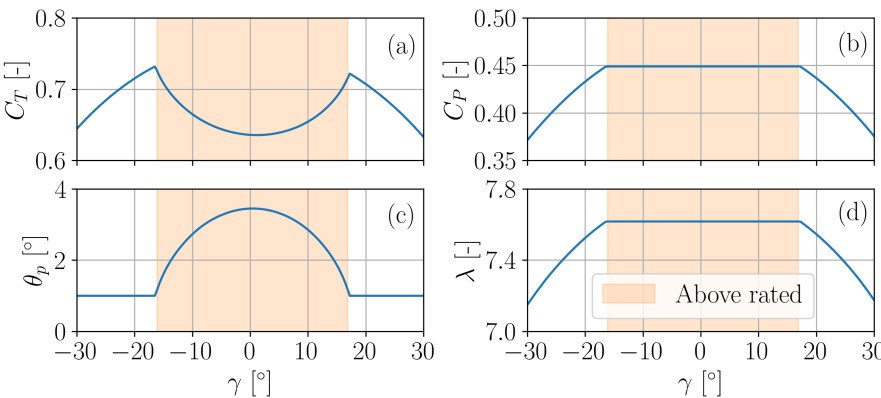

**Figure 6.** Behavior of the IEA 3.4 MW reference turbine as it yaws out of the wind, transitioning between the above-rated region III and the below-rated region II. Thrust coefficient $C_T$ **(a)**, power coefficient $C_P$ **(b)**, blade pitch angle $\theta_p$ **(c)**, and tip speed ratio $\lambda$ **(d)**. An interactive version of the figure is available as a Jupyter notebook at the link https://tinyurl.com/btcl-fig-6.

Differently from regions II and III, where controllers only require knowledge of the rotor speed, the control policy in region II1/2 typically requires prescribing the desired pitch and torque settings as functions of wind speed. In other words, one has to provide the schedules $\theta_p = \theta_p(u_\infty)$ and $Q_a = Q_a(u_\infty)$ in the desired range $u_\infty^{lb} \leq u_\infty \leq u_\infty^{ub}$. Two common examples of region II1/2 control policies are provided by load and noise alleviation techniques.

     Load alleviation is often necessary because thrust reaches a sharp maximum at rated wind speed, $T_r = \frac{1}{2}\rho A u_{\infty_r}^2 C_T(\theta_p^*, \lambda^*)$.
To reduce the effects of this large load on the sizing of various turbine components, thrust clipping (or peak shaving) is used, where blades are pitched to feather according to a desired schedule $\theta_p = \theta_p(u_\infty)$. This has the effect of reducing the angle of attack and hence the thrust (Zalkind et al., 2022), at the price of some reduced power. To minimize power losses for a given pitch schedule, the optimal power coefficient schedule can be computed as $C_P(u_\infty) = \max_\Omega C_P(\Omega R/u_\infty, \theta_p(u_\infty))$ under the constraint $\Omega \leq \Omega_r$, which also returns the rotor speed schedule $\Omega(u_\infty)$. Consequently, the torque schedule becomes
$Q_a(u_\infty) = \frac{1}{2}\rho A u_\infty^3 C_P(u_\infty)/\Omega(u_\infty)$.

     One effective way of constraining noise emissions is to limit the rotor speed (Leloudas et al., 2007; Bottasso et al., 2012) to a maximum noise-acceptable value $\Omega_n$. When $\Omega_n < \Omega_r$, the increase in rotor speed as a function of wind speed that characterizes region II is stopped before the machine reaches rated power. In this case, region II1/2 is entered when $u_\infty > u_\infty^{lb} = \Omega_n R/\lambda^*$ and, above this wind speed, the rotor operates at the constant speed $\Omega_n$. To minimize power losses, the blade pitch setting can
be computed for each wind speed $u_\infty$ as $\theta_p(u_\infty) = \arg\max_{\theta_p} C_P(\lambda(u_\infty), \theta_p)$, where $\lambda(u_\infty) = \Omega_n R/u_\infty$. The corresponding aerodynamic torque schedule is readily obtained as $Q_a(u_\infty) = \frac{1}{2}\rho A R C_P(\lambda(u_\infty), \theta_p(u_\infty))/\lambda(u_\infty)$. The end of region II1/2 is reached when, for sufficiently high $u_\infty$, the turbine reaches rated power, finally entering into region III.

     In order to implement a given control policy for region II1/2 with the proposed model, we assume that the controller will implement the desired schedules $\theta_p(u_\infty)$ and $Q_a(u_\infty)$ (whether computed as explained above or according to different criteria)

by reacting to the rotor-orthogonal wind speed component $u_\infty \cos\gamma$. In the absence of specific details on the implementation, this is a reasonable assumption, as region II1/2 controllers are typically based on rotor-effective wind speed estimates (Bottasso et al., 2012; Zalkind et al., 2022) that, in a misaligned condition, will sense $u_\infty \cos\gamma$ and not $u_\infty$. Therefore, when $u_\infty^{lb} \leq u_\infty \cos\gamma \leq u_\infty^{ub}$, Eq. (30) is solved by appending to it the following two constraints:

$$\theta_p = \theta_p(u_\infty \cos\gamma), \tag{33a}$$

$$Q_a = Q_a(u_\infty \cos\gamma). \tag{33b}$$

In summary, using Eq. (30) in combination with the constraint Eqs. (31), (32), or (33), yields the setpoint achieved by the turbine for given ambient conditions and a given misalignment, no matter what region it corresponds to and what control strategy is implemented by the controller.

### 3.4 Model calibration

Through Eq. (25b) and (26), the analytical model depends on $C_D$, $C_{L,\alpha}$ and $\beta$. These are average parameters, which represent in the model the "equivalent" effect caused by corresponding quantities that in reality exhibit a spanwise variability.

When numerical or experimental measurements are available, the parameters $C_D$, $C_{L,\alpha}$, and $\beta$ can be calibrated to minimize the error produced by the model in the prediction of the power loss factor $\eta_P$ and of the thrust coefficient $C_T$. Notice that $C_T$ is preferred to $\eta_T$ for this scope, because it was found that the informational content of $\eta_T$ is very similar to the one of $\eta_P$, reducing the quality of the tuning. Calibration is here performed by numerically solving the following minimization problem

$$\min_{C_D, C_{L,\alpha}, \beta} \sqrt{\frac{1}{N} \sum_{i=1}^{N} \left(\eta_{P,i}^{\text{obs}} - \eta_{P,i}^{\text{mod}}\right)^2} + \sqrt{\frac{1}{N} \sum_{i=1}^{N} \left(C_{T,i}^{\text{obs}} - C_{T,i}^{\text{mod}}\right)^2}, \tag{34}$$

where $(\cdot)_i^{\text{obs}}$ are $N$ numerical or experimental observations, and $(\cdot)_i^{\text{mod}}$ the corresponding model predictions. For each data set, tuning was performed solving $N/2$ times the problem expressed by Eq. (34) using a gradient based optimization, each time with a different random 50% subset of the available data, and finally averaging the resulting parameters.

### 3.5 Simplified choice of model parameters in the absence of calibration data

When calibration is not possible, the equivalent model parameters $C_D$, $C_{L,\alpha}$ and $\beta$ must be estimated from the corresponding actual spanwise distributions $C_D(r/R)$, $C_{L,\alpha}(r/R)$ and $\beta(r/R)$ .

Examining the expression for thrust given by Eq. (16), neglecting drag, it appears that the lift force has roughly a spanwise triangular distribution. In fact, inspecting Eqs. (5), $u_t$ is proportional to $r$, whereas $u_n$ does not depend on $r$. Additionally, the leading term of the Taylor series of the optimal twist distribution is $1/r$ (Burton et al., 2011). Similarly, inspecting the expression for power given by Eq. (21), again neglecting the contribution of drag, it appears that – for the same reasons – also the spanwise power capture has a triangular distribution. This suggests to evaluate the spanwise integrals at the centroid of the

triangle, which is located at $r/R = 2/3$. Adopting this approach, the model parameters are then set to the following values:

$$C_D = f_d C_D(2/3), \tag{35a}$$

$$C_{L,\alpha} = f_l C_{L,\alpha}(2/3), \tag{35b}$$

$$\beta = \beta(2/3). \tag{35c}$$

Coefficient $f_d$ is a correction factor for drag, while $f_l$ is a knockdown factor for lift, which accounts for the finite span of the blades. Based on comparisons with calibrated values (see Sect. 4.2), we recommend a drag correction factor $f_d = 1$ for moderate yaw (up to 20°) and pitch values, and a smaller value of 0.45 if the model has to be used also for large yaw and pitch settings. This smaller value is probably due to the approximation of a small inflow angle used in the model, which is partially corrected by a smaller drag coefficient. For lift, we recommend the value $f_l = 2/3$.

The performance of this simplified choice of model parameters is demonstrated later in Sect. 4.2.

## 4  Model validation

### 4.1  Validation with respect to LES-ALM simulations

LES-ALM simulations are used for testing the accuracy of the model in representing misaligned conditions, similarly to what done by other authors (Gebraad et al., 2016; Liew et al., 2020; Nanos et al., 2022). The effects of the rotor on the flow are modelled with the filtered ALM of Troldborg et al. (2007) and Martínez-Tossas and Meneveau (2019), by projecting forces computed along the lifting lines onto the LES grid. The Cartesian mesh consists of approximately 3.5 million cells, and uses four refinement levels. The smallest cells measure 1 m, and are located in correspondence of the rotor.

Simulations were conducted for the IEA 3.4 MW reference wind turbine, whose complete technical specifications are reported in Bortolotti et al. (2019). Here we only note that the turbine has a 5° uptilt angle, i.e. $\delta = -5°$. The parameters of the proposed model were calibrated as explained in Sect. 3 based on the LES-ALM simulations described in Sect. 4.1.2, but not the ones of Sect. 4.1.1, obtaining the values $C_D = 0.0052 \pm 0.0001$, $C_{L,\alpha} = 4.759 \pm 0.007$ rad$^{-1}$, and $\beta = -3.345 \pm 0.007°$, for a 95% confidence level.

### 4.1.1  Simulations in control regions II and III

First we demonstrate the integration with a standard controller, including operations in region II, III, and the transition between the two as the turbine is progressively yawed out of the wind. For the LES-ALM simulations, setpoints were computed using a controller in the loop, based on an implementation similar to the one of Bortolotti et al. (2019). For the proposed model, setpoints were obtained from Eq. (30) in combination with the constraint Eqs. (31) and (32).

We consider laminar inflows with four wind speeds: one below-rated speed of 8.5 ms$^{-1}$, and three above-rated speeds of 10.5, 11, and 13 ms$^{-1}$. Figure 7 reports the results in term of the thrust loss factor $\eta_T$ (panel a), power loss factor $\eta_P$ (panel b),

blade pitch $\theta_p$ (panel c), and tip speed ratio (panel d), all plotted as functions of the misalignment angle $\gamma$. The solid markers are the results of the LES-ALM simulations, while the lines represent predictions of the proposed model.

For the lowest wind speed of 8.5 ms$^{-1}$, the turbine always operates in region II. This is the only case where the method of Heck et al. (2023) is strictly applicable. In fact, their method does not contain a generic thrust model, but rather it is formulated in terms of the modified thrust coefficient $C'_T$, which is constant when a turbine yaws out of the wind in region II. The results of this alternative model are shown with a dashed orange line in the figure. The benchmark LES simulations feature a negative uptilt $\delta = -5°$, which is not modelled in the approach of Heck et al. (2023). To avoid cluttering the results with this additional

effect, here and in the following examples the total true misalignment $\mu$ (instead of $\gamma$) is provided as input to the model of Heck et al. (2023). The figure shows that both methods are in excellent agreement with the CFD results.

     For the highest wind speed of 13 ms$^{-1}$, the turbine operates in region III for all misalignment angles. On the other hand, for a wind speed of 11 ms$^{-1}$ the machine enters region II around $\gamma = 27°$, and for 10.5 ms$^{-1}$ at about $\gamma = 16°$.

     In general, there is a very good agreement of the model with the higher fidelity CFD results, not only in terms of loss factors,

but also on the calculation of the setpoints.

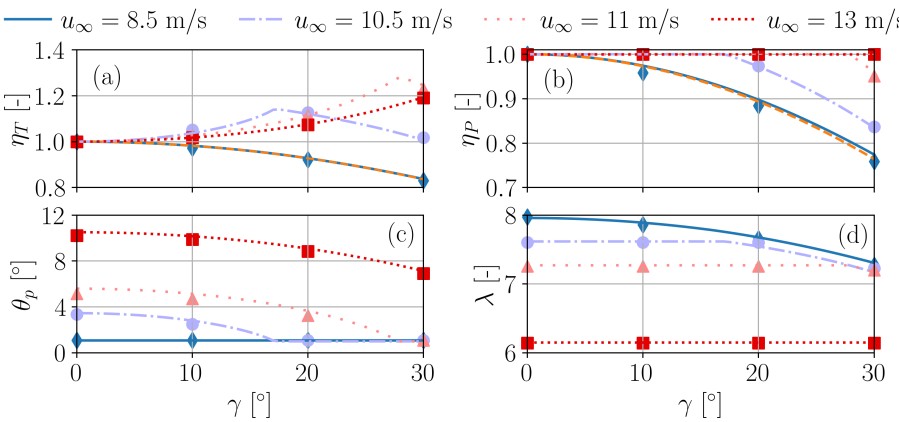

**Figure 7.** Thrust loss factor $\eta_T$ **(a)**, power loss factor $\eta_P$ **(b)**, blade pitch $\theta_p$ **(c)**, and tip speed ratio $\lambda$ **(d)**, plotted as functions of the misalignment angle $\gamma$, for various below, above and around rated wind speeds. Proposed model: lines; LES-ALM: solid markers; Heck et al. (2023) (only for the 8.5 ms$^{-1}$ case): dashed orange lines.

### 4.1.2   Simulations with fixed tip speed ratio and pitch setting

Next, we present a second set of results obtained by varying the misalignment while keeping the tip speed ratio and pitch constant, for different inflows. These conditions are meant to provide a more general view of the performance of the proposed method in a variety of conditions, although – as explained in Sect. 3.3 – tip speed ratio and pitch in general do not both remain

constant when a turbine yaws out of the wind.

The four operational scenarios of Table 1 are considered. The flow is laminar and steady in all scenarios. Cases 1 and 2 have no shear and different tip speed ratios, whereas cases 3 and 4 are sheared and have the same $\lambda$.

**Table 1.** Operational scenarios for the LES-ALM simulations.

| Scenario # | 1 | 2 | 3 | 4 |
|---|---|---|---|---|
| $\lambda$ [-] | 8 | 9.5 | 8.38 | 8.38 |
| $k$ [-] | 0 | 0 | 0.06 | 0.19 |

Figures 8 and 9 report the power loss factor $\eta_P$ in the range of yaw misalignment angles $-30° < \gamma < 30°$ for different pitch settings, each corresponding to a different thrust coefficient $C_{T,0}$ in aligned conditions. Figure 8 corresponds to scenarios 1
490 and 2 of Table 1, i.e. no shear, while Fig. 9 reports the solution for the sheared cases 3 and 4 of that same table.

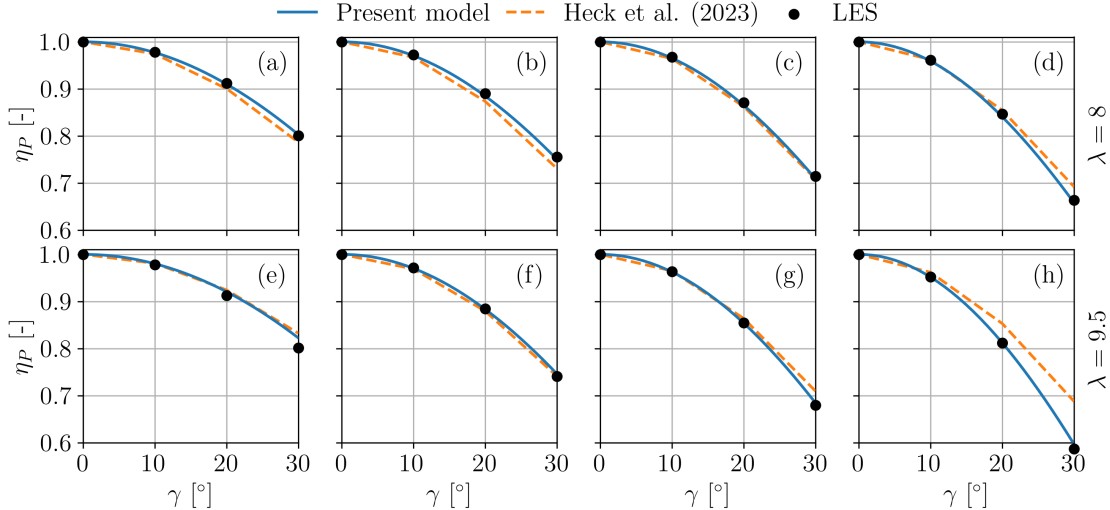

**Figure 8.** Power loss factor $\eta_P$ vs. misalignment angle $\gamma$ in the unsheared scenarios 1 ($\lambda = 8$) **(a-d)** and 2 ($\lambda = 9.5$) **(e-h)**. Each subplot corresponds to a different value of the thrust coefficient in wind-aligned conditions: $C_{T,0} = 0.74$ **(a)**; $C_{T,0} = 0.54$ **(b)**; $C_{T,0} = 0.44$ **(c)**; $C_{T,0} = 0.36$ **(d)**, $C_{T,0} = 0.86$ **(e)**; $C_{T,0} = 0.60$ **(f)**; $C_{T,0} = 0.47$ **(g)**; $C_{T,0} = 0.35$ **(h)**.

In the figures, LES-ALM results (shown with black circular markers) are compared with the proposed approach (shown with blue solid lines) and the method proposed by Heck et al. (2023) (shown with orange dashed lines). For the latter, the modified thrust coefficient $C_T'$ was obtained directly from each LES simulation at the corresponding $\gamma$ value.

  Overall, there is a very good match between the predictions of the proposed model and numerical simulations. For the null
495 shear cases of Fig. 8, results are reported only for positive yaw angles, as power is symmetric. On the other hand, power is not symmetric for the sheared inflow cases of Fig. 9, which shows clear evidence of the complex behavior described in Sect. 2.8. At

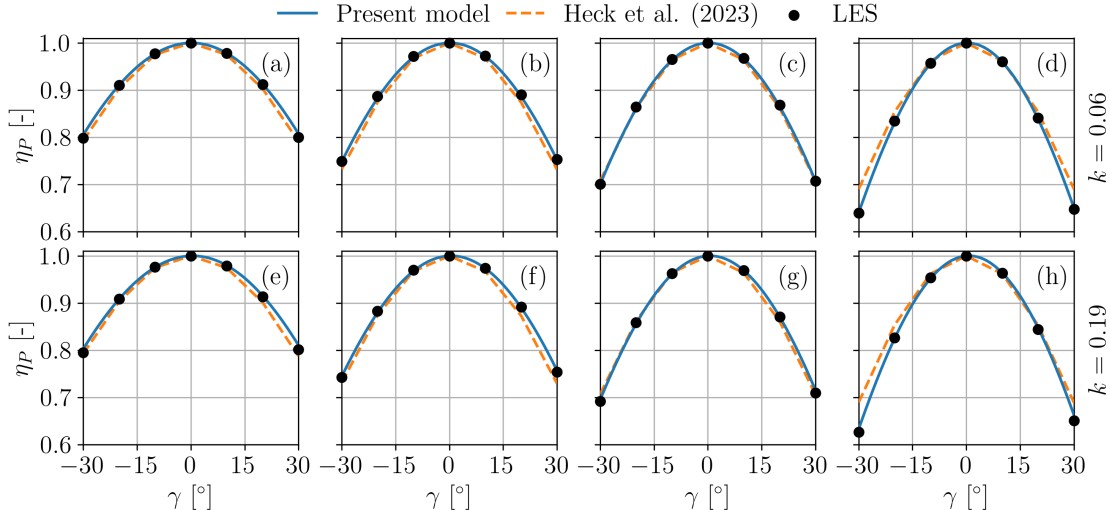

**Figure 9.** Power loss factor $\eta_P$ vs. misalignment angle $\gamma$ in the sheared scenarios 3 ($k = 0.06$) **(a-d)** and 4 ($k = 0.19$) **(e-h)**. Each subplot corresponds to a different value of the thrust coefficient in wind-aligned conditions: $C_{T,0} = 0.77$ **(a)**; $C_{T,0} = 0.56$ **(b)**; $C_{T,0} = 0.45$ **(c)**; $C_{T,0} = 0.36$ **(d)**, $C_{T,0} = 0.77$ **(e)**; $C_{T,0} = 0.56$ **(f)**; $C_{T,0} = 0.45$ **(g)**; $C_{T,0} = 0.36$ **(h)**.

high $C_T$ (low pitch), curves are very nearly symmetric with respect to $\gamma$. However, as thrust is decreased (and pitch increased), power capture is larger at positive than negative $\gamma$ values.

The model of Heck et al. (2023) performs similarly well at high and moderate rotor loading, when $C_T'$ is roughly constant.
However, as the $C_T$ is reduced, the model tends towards the solution $\cos^3 \gamma$, and therefore its accuracy is compromised. Moreover, the model fails to predict the shear-induced asymmetry (see Fig. 9).

As predicted by the proposed model, the power asymmetry increases with shear (see Table 1 and the explanation given in Sect. 2.8). To facilitate the visualization of this effect, Fig. 10 shows the difference $\Delta\eta_{P,\gamma=\pm30°}$ between the two values of $\eta_P$ at $\gamma = \pm30°$ as a function of shear, for varying thrust coefficients. The asymmetry exhibits also a noticeable dependency on the
505 thrust coefficient, larger asymmetries being observed for lower values of $C_T$, as predicted in Sect. 2.8 by examining Eq. (22).

Figures 11 and 12 report the thrust loss factor $\eta_T$ as a function of yaw misalignment, for the same four scenarios and different thrust settings. Here again, model predictions are indicated with lines, and LES-ALM results with markers. There is a consistently good match, for all scenarios, and for all yaw and pitch values. The lack of symmetry is again consistent with the model, similarly to the case of power discussed above. Figures 12a to 12d show a higher thrust for positive yaw angles at low
thrust coefficients (high pitch values), because of the high tip speed ratio of scenario 3, indicating that term $C_{T_2}$ prevails over $C_{T_1}$. The opposite happens in Fig. 12e to 12h, due to the lower $\lambda$ of scenario 4.

Overall, it appears that the performance of the rotor is strongly dependent on thrust coefficient and tip speed ratio, and hence on the way it is controlled when it yaws out of the wind. Therefore, the standard power law $\cos^{p_P}$ may oversimplify the

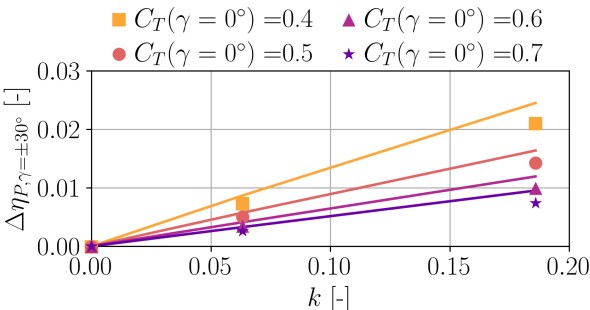

**Figure 10.** Difference $\Delta\eta_{P,\gamma=\pm30°}$ between power loss factors $\eta_P$ evaluated at misalignments $\gamma = 30°$ and at $\gamma = -30°$, as a function of vertical linear shear coefficient $k$, for varying thrust coefficient $C_T$. Proposed model: solid lines; LES-ALM simulations: markers.

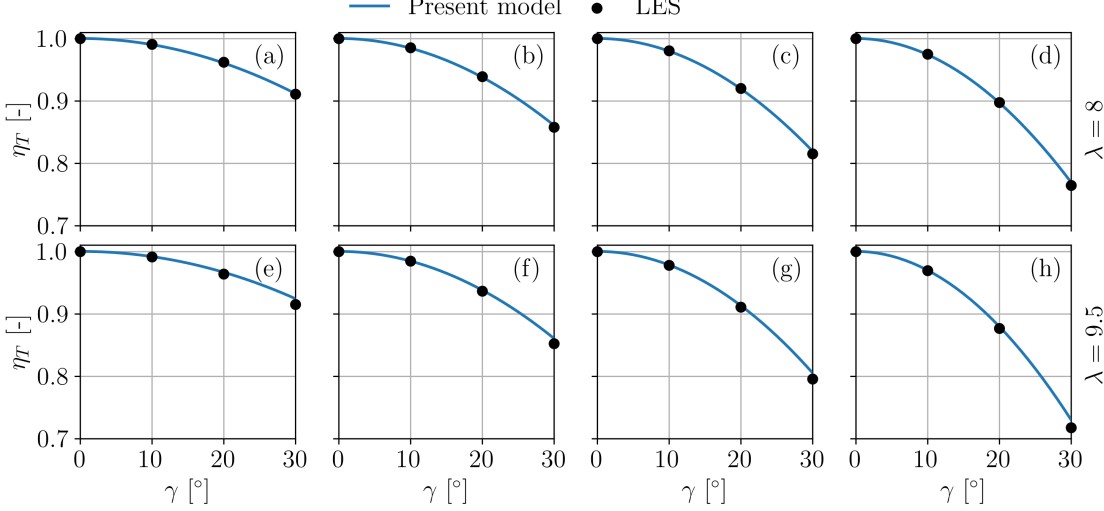

**Figure 11.** Thrust loss factor $\eta_T$ vs. misalignment angle $\gamma$ in scenario 1 with $\lambda = 8$ **(a-d)** and scenario 2 with $\lambda = 9.5$ **(e-h)**. Each subplot corresponds to a different value of the thrust coefficient in wind-aligned conditions: $C_{T,0} = 0.74$ **(a)**; $C_{T,0} = 0.54$ **(b)**; $C_{T,0} = 0.44$ **(c)**; $C_{T,0} = 0.36$ **(d)**, $C_{T,0} = 0.86$ **(e)**; $C_{T,0} = 0.60$ **(f)**; $C_{T,0} = 0.47$ **(g)**; $C_{T,0} = 0.35$ **(h)**.

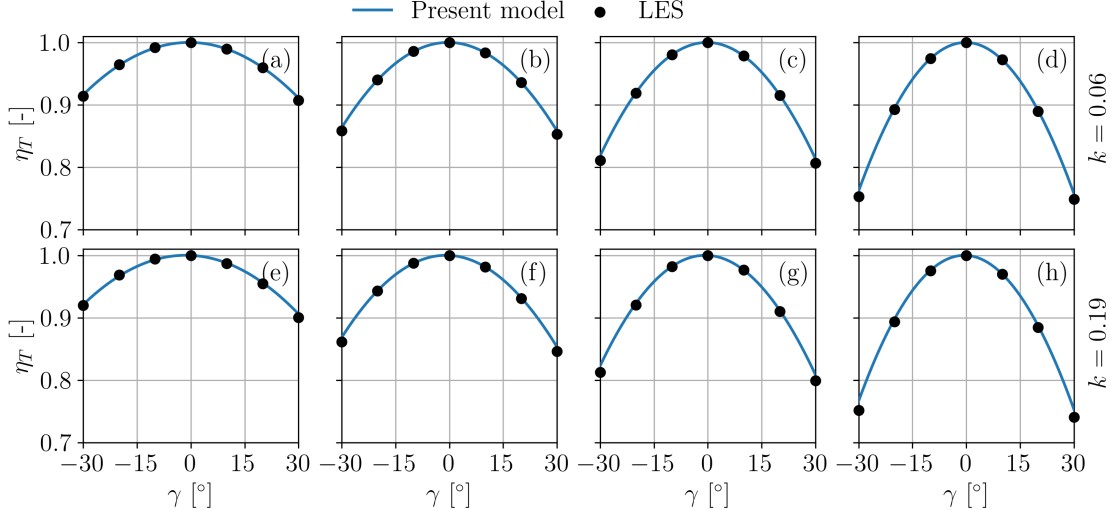

**Figure 12.** Thrust loss factor $\eta_T$ vs. misalignment angle $\gamma$ in scenarios 3 with $k = 0.06$ **(a-d)** and scenario 4 with $k = 0.19$ **(e-h)**. Each subplot corresponds to a different value of the thrust coefficient in wind-aligned conditions: $C_{T,0} = 0.77$ **(a)**; $C_{T,0} = 0.56$ **(b)**; $C_{T,0} = 0.45$ **(c)**; $C_{T,0} = 0.36$ **(d)**, $C_{T,0} = 0.77$ **(e)**; $C_{T,0} = 0.56$ **(f)**; $C_{T,0} = 0.45$ **(g)**; $C_{T,0} = 0.36$ **(h)**.

complex aerodynamics that are typical of this problem. On the other hand, notwithstanding its simplicity, the proposed model

is in very good agreement with sophisticated CFD simulations, and it is capable of describing even relatively minor effects of the complex behavior of a misaligned wind turbine rotor in a sheared inflow.

### 4.2 Validation of the simplified choice of model parameters

The simplified choice of model parameters described in Sect. 3.5 is based on Eqs. (35), which include the correction factors $f_d$ and $f_l$. To verify the existence of typical values for these factors, we considered four different wind turbines: IEA 3.4 MW

(Bortolotti et al., 2019); NREL 5 MW (Jonkman et al., 2009); G178, which is a modified version of the DTU 10 MW (Bak et al., 2013; Wang et al., 2021); and the small-scale G1 turbine (Bottasso and Campagnolo, 2022; Campagnolo et al., 2020). For the three full-scale machines, the model parameters were first calibrated using the LES-ALM simulation results of Sect. 4.1.2, whereas for the G1 model the calibration was performed using wind tunnel measurements (see later Sect. 4.3).

The parameters calibrated this way were then compared to the ones based on the simplified approach of Eqs. (35), leading

to the recommended values reported in Sect. 3.5. The $C_D(r/R)$ and $C_{L,\alpha}(r/R)$ coefficients were obtained by averaging over the interval of angles of attack $2°$ below the negative and positive stall limits. In all cases, the calibrated value of the twist corresponded remarkably well with the actual twist at $2/3$ span, i.e. $\beta(2/3)$.

The simplified choice of the model parameters was then applied to the NREL 5 MW and G178 10 MW wind turbines. Simulations were performed with a steady inflow, at different misalignments and for two different blade pitch settings. The

530 proposed model was calculated with the parameters based on Eqs. (35), using the default correction coefficients $f_d = 0.45$ and $f_l = 2/3$ (in other words, without using LES-ALM calibrated values, replicating what one could do in the absence of suitable tuning data).

The results in terms of $\eta_T$ and $\eta_P$ for the two turbines are reported in Fig. 13, and compared with LES-ALM simulations. For both turbines there is an excellent match between model predictions and CFD results. This seems to indicate that the even 535 a simplified choice of the model parameters is sufficient for a good performance of the model.

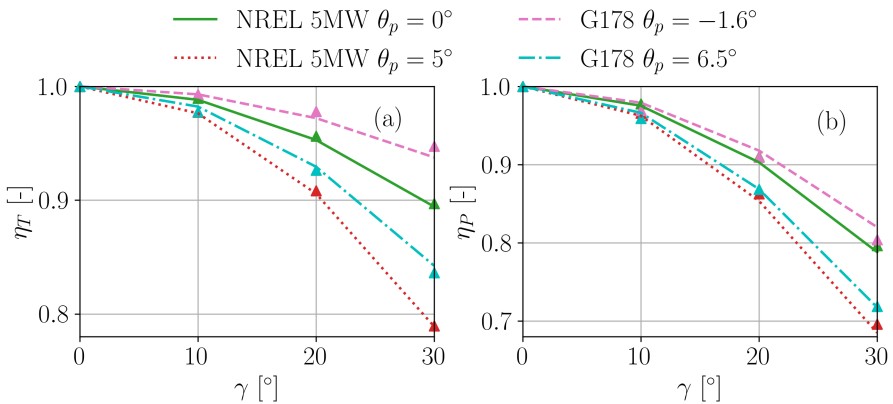

**Figure 13.** Thrust $\eta_T$ **(a)** and power $\eta_P$ **(b)** loss factors as functions of the misalignment angle for two pitch settings. Results obtained with the NREL 5 MW (Jonkman et al., 2009) and G178 10 MW (Bak et al., 2013; Wang et al., 2021) wind turbines, using the simplified calculation of the model parameters based on Eqs. (35). Proposed model: lines; LES-ALM simulations: markers.

### 4.3 Validation with respect to wind tunnel measurements

Next, the model is compared to data recorded during wind tunnel experimental campaigns performed with a G1 wind turbine (Campagnolo et al., 2016). This scaled machine has a diameter of 1.1 m, a rated rotor speed of 850 RPM, and null tilt. The design of the G1 is described in Bottasso and Campagnolo (2022), and its rotor aerodynamic and wake characteristics have 540 been reported in Wang et al. (2021) and references therein.

Tests were performed in a boundary layer wind tunnel (Bottasso et al., 2014) with three different inflows: the first one, termed Low-TI, has no shear and a very low turbulence intensity (approximatively equal to 1%); the other two, termed Mod-TI and High-TI, have respectively TIs of about 6% and 13% at hub height, and vertical linear shears in the rotor region equal to $k = 0.11$ and $k = 0.15$, respectively. Figure 14a reports the vertical profiles of the longitudinal wind speed component $u$ 545 measured by means of CTA probes (Bottasso et al., 2014), normalized by the wind speed $u_{\text{pitot}}$ measured by a Pitot tube placed at hub height. Figure 14b shows the vertical profiles of the turbulence intensity, as measured with the same instrumentation.

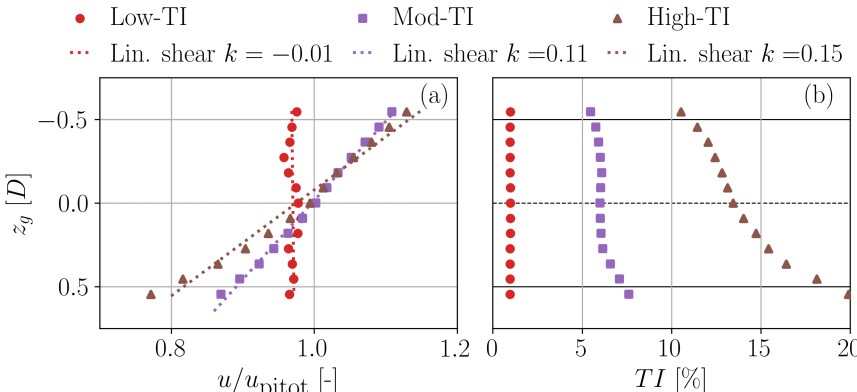

**Figure 14.** Wind tunnel inflows. Vertical wind speed profiles, with corresponding best-fitted linear shears (dotted lines) **(a)**; vertical profiles of turbulence intensity **(b)**.

The experimental characterization of power losses in misaligned conditions was performed based on the three campaigns of Table 2, totalling 119 observations of a duration of 2 minutes each. The measured average $\theta_p$ and $\lambda$ are reported in Appendix C for campaigns 1 and 2, and in Fig. 19 for campaign 3.

The first campaign was conducted in region II (i.e., the below-rated partial-load regime), using the classical variable-rotor-speed maximum-power tracking strategy. Tests were conducted in all three Low-TI, Mod-TI, and High-TI inflow conditions, with hub-height wind speeds of 5.86, 5.69, and 5.40 ms$^{-1}$, respectively.

    The second campaign was also conducted in region II, but in this case the turbine was derated in the range $P_d \in [50, 100]\%$ while adopting two different strategies: iso-$\lambda$, where the tip speed ratio is held constant (Campagnolo et al., 2023), and min-

$C_T$, where the thrust coefficient is minimized (Juangarcia et al., 2018). Tests were conducted only in the Mod-TI inflow, with a hub-height wind speed of 5.62 ms$^{-1}$,

    The third campaign was conducted in region III (i.e., the above-rated full-load regime) in the Low-TI and Mod-TI inflows, with hub-height wind speeds of 6.97 and 6.11 ms$^{-1}$, respectively.

**Table 2.** Characteristics of the wind tunnel campaigns.

| Campaign # | Inflow wind speed | | | Control region | Used for tuning |
| --- | --- | --- | --- | --- | --- |
| | Low-TI | Mod-TI | High-TI | | |
| 1 - Max $P$ tracking | 5.86 ms$^{-1}$ | 5.69 ms$^{-1}$ | 5.40 ms$^{-1}$ | II | Yes |
| 2 - Derating | - | 5.62 ms$^{-1}$ | - | II | Yes |
| 3 - Above rated | 6.97 ms$^{-1}$ | 6.11 ms$^{-1}$ | - | III | No |

    Various sources of error affect the experimental observations. These include measurements of the wind speed $u_{\text{pitot}}$ upstream

of the model (obtained by a Pitot tube placed at hub height 3D in front of the turbine), of the air density $\rho$, of the rotor speed

$\Omega$, of the shaft torque $Q$, of the bending moment at tower base (which is used to estimate thrust), of the blade pitch angle, and of the nacelle orientation with respect to the wind tunnel (i.e., of the misalignment angle). The error in $u_{\text{pitot}}$ is related to the uncertainty associated with the measurements of flow density and dynamic pressure. This latter quantity is measured with a MKS Baratron-Type 226A transducer (MKS Instruments, Inc., 2022) with full span equal to 1 Torr, characterized by an accuracy of $\pm 0.4$ Pa. Density is instead derived from measurements of air pressure, temperature and humidity, and it is affected by an error equal to $\pm 0.01$ kgm$^3$ (Wang et al., 2020). Torque is measured with a load cell installed on the rotor shaft, and it is affected by an uncertainty of $\pm 0.005$ Nm. The rotor speed measurement, provided by an optical incremental encoder, is instead affected by an error equal to $\pm 1.5$ RPM. The measurement uncertainty on power $P = Q\Omega$ is derived by adding in quadrature the uncertainties on $Q$ and $\Omega$. Thrust $T$ is obtained by correcting the measurements of the bending moments at tower base by the effects induced by the drag of the tower, nacelle and hub spinner (Wang et al., 2020). The calibration of the load cell at tower base revealed an uncertainty in the thrust of $\pm 0.14$ N. Blade pitch and nacelle orientation are measured by optical encoders, affected by uncertainties of $\pm 0.2°$. In turn, all these effects are used to quantify uncertainties in the tip speed ratio $\lambda$, and yaw-induced power and thrust losses $\eta_P$ and $\eta_T$, again by adding errors in quadrature. In the following, the resulting uncertainties are reported for a 95% confidence level.

Uncertainties in some experimental measurements affect also the predictions of the proposed model. The uncertainties of the four model inputs – tip speed ratio, blade pitch, rotor speed, and yaw misalignment – were propagated forward throughout the model by Latin hypercube sampling with ten thousand sample points, using the UQLab software (Marelli and Sudret, 2014).

Tuning of the model parameters $C_D$, $C_{L,\alpha}$ and $\beta$ was performed with Eq. (34), using a total of 94 observations from the experimental campaigns 1 and 2.

Given the small size of the G1 wind turbine, the Reynolds number at its blade sections is particularly low. Although special low-Reynolds airfoils are used in the design of the G1 blades (Bottasso and Campagnolo, 2022), their aerodynamic characteristics are particularly sensitive to the operating conditions of the turbine (Wang et al., 2020). In fact, the Reynolds number has a significant effect on the drag and on the zero-lift direction, which in turn affects the parameter $\beta$, whereas the effect on the lift slope $C_{L,\alpha}$ is negligible (Wang et al., 2020). Accordingly, the model parameters $C_D$ and $\beta$ are assumed to depend on the rotational speed $\Omega$, since the relative speed at the airfoils is close to the tangential speed ($u \approx u_t$, see Fig. 3). The values of $C_D$ and $\beta$ at $\Omega = [850, 625, 400]$ RPM were assumed as unknowns, and a piecewise linear interpolation was used at other intermediate values of the rotor speed.

Figure 15 reports the tuned $C_D$ and $\beta$ parameters, whiskers representing the corresponding 95% confidence intervals. As expected, drag decreases for increasing rotor speed, i.e. for increasing sectional Reynolds number. The twist $\beta$ also exhibits the same trend, since the zero-lift direction rotates nose-up as the Reynolds number increases (Wang et al., 2020). The tuned parameter $C_{L,\alpha}$ is equal to $4.5033 \pm 0.0459$.

For maximum power tracking operation in region II (test campaign 1), Fig. 16 reports a comparison between model-predicted and measured power and thrust losses. The present model results are indicated with blue solid lines, while measurements are indicated by black circles. Whiskers indicate the respective 95% confidence intervals. There is a very good match between experimental measurements and the present model, the latter falling within the uncertainty range of the measurements in most

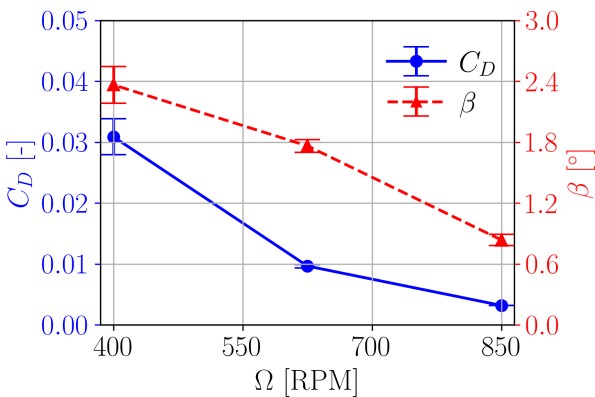

**Figure 15.** Tuned model parameters $C_D$ and $\beta$ as functions of rotational speed $\Omega$.

cases. As predicted by the model, the sheared inflow conditions Mod-TI and High-TI exhibit the expected non-symmetric behavior with respect to positive and negative yaw angles.

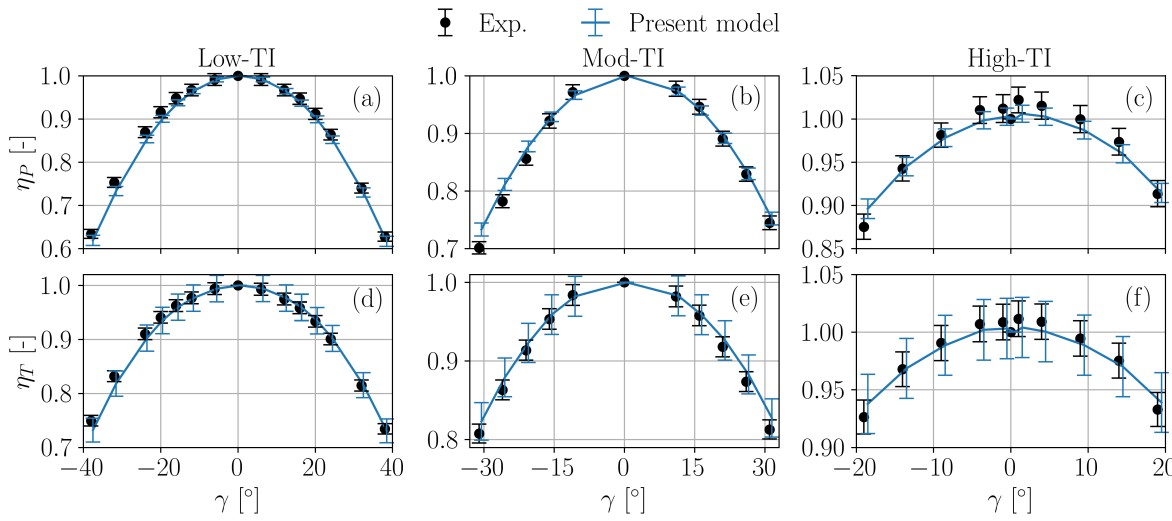

**Figure 16.** Experimental campaign 1. Power loss factor $\eta_P$ **(a-c)** and thrust loss factor $\eta_T$ **(d-f)** vs. yaw misalignment $\gamma$, in region II operation for the three different inflow conditions. Whiskers indicate the 95% confidence intervals.

In principle, the model of Heck et al. (2023) would be applicable to these tests in region II. However, their method does not directly consider the aerodynamic characteristics of the blades, as it expresses their behavior through the single parameter represented by $C'_T$. Therefore, it is blind to the variability of twist and drag with respect to the Reynolds number, which drops significantly as the misalignment angle increases. Since this strong Reynolds-dependency is specific to the small scale

of wind tunnel models, the results of the method of Heck et al. (2023) are not shown here, because its poor match with the measurements would be misleading, as these effects would not be present at full scale. The present method is not affected by this issue, because it uses a lifting line approach and specifically includes the blade aerodynamic characteristics in the governing Eqs. (22) and (25b).

The loss factors are reported for derated operation (test campaign 2) in Fig. 17. Here again the match between experimental data and predictions of the present model is very good, the latter being mostly within the uncertainty band of the measurements. Slightly larger deviations are observed for the min-$C_T$ case at $P_d$=50%. This can be explained by the fact that the machine operates at significantly low $\lambda$ values, with consequent low rotational speeds. This results in particularly high angles of attack (Juangarcia et al., 2018) and very low chord-based Reynolds. Both have significant impacts on the airfoil performance, which are likely not properly captured by the analytical model. Overall, it appears that the model is capable of capturing the reduction in the thrust coefficient as derating $P_d$ increases, as well as the lack of symmetry with respect to the misalignment angle.

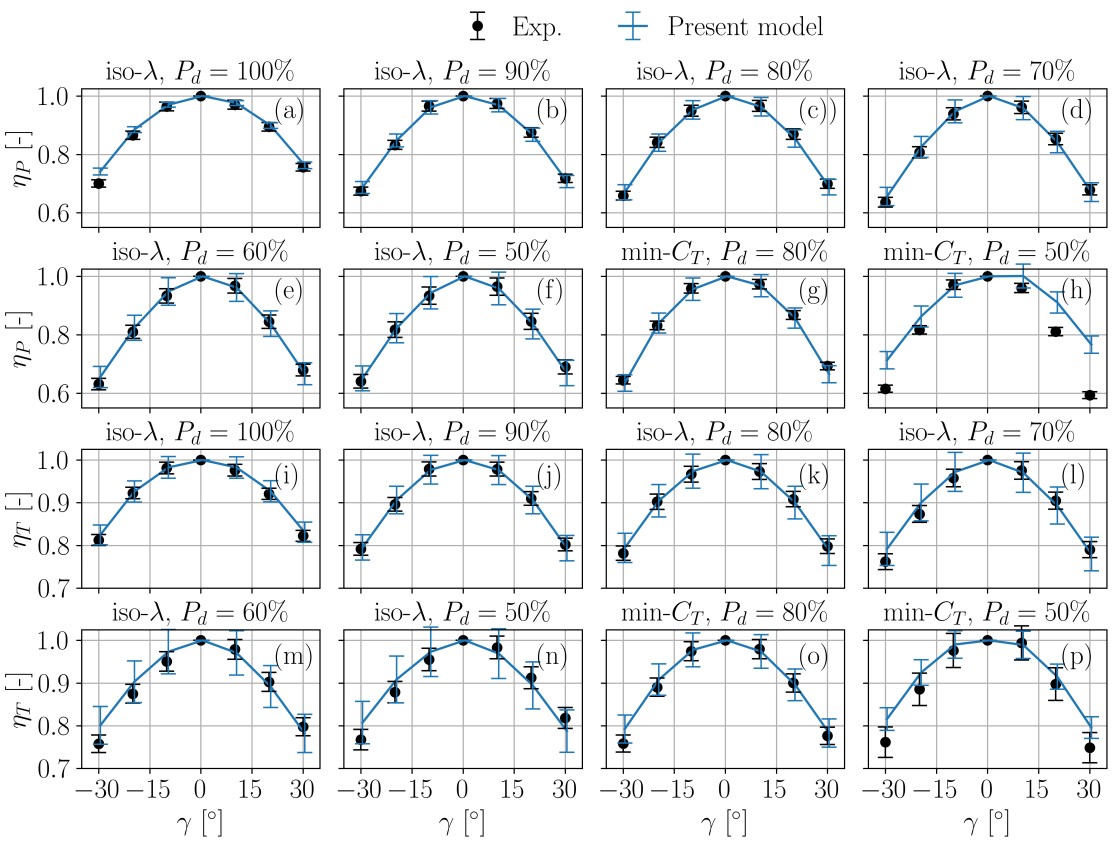

**Figure 17.** Experimental campaign 2. Power loss factor $\eta_P$ **(a-h)** and thrust loss factor $\eta_T$ **(i-p)** vs. yaw misalignment $\gamma$, in derated operation in region II for the Mod-TI inflow case. Whiskers indicate the 95% confidence intervals.

The effects of thrust and shear are visualized in Fig. 18 in terms of the average and the difference of the power loss factors at $\gamma \pm 20°$, respectively noted $\bar{\eta}_{P,\gamma=\pm20°}$ and $\Delta\eta_{P,\gamma=\pm20°}$. It appears that power losses tend to decrease with increasing thrust coefficients (panel a), whereas there is no significant dependency on shear (panel b). The power loss asymmetry grows with increasing shear (panel d). On the other hand, the asymmetry is roughly constant with respect to the thrust coefficient (panel c).

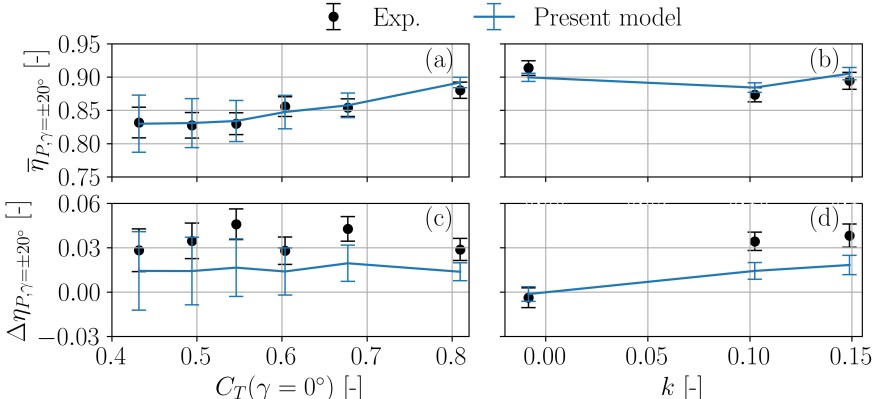

**Figure 18.** Power loss factor $\bar{\eta}_{P,\gamma=\pm20°}$ (averaged between $\gamma = \pm20°$) as a function of $C_T(\gamma = 0°)$ **(a)** and of shear $k$ **(b)**. Difference $\Delta\eta_{P,\gamma=\pm20°}$ between power loss factors $\eta_P$ evaluated at misalignments $\gamma = 20°$ and at $\gamma = -20°$, as a function of $C_T(\gamma = 0°)$ **(c)** and of shear $k$ **(d)**. Proposed model: solid lines; experimental measurements: black circles. Whiskers indicate the 95% confidence intervals.

In the third test campaign, the wind turbine is operated above rated conditions. Figure 19 reports the 2-minute average tip speed ratio and pitch angles measured during the experiment, and plotted as functions of the misalignment angle $\gamma$. For the Low-TI case, the turbine operates in region III for all misalignment angles. Recalling that the tip speed ratio is defined as $\lambda = \Omega R / u_{\infty,\text{hub}}$, since both the ambient wind speed $u_{\infty,\text{hub}}$ and rotor speed $\Omega$ are constant, when the turbine yaws away from the wind $\lambda$ (indicated by red circles in Fig. 19a) remains constant, while the blades are pitched back (red circles in Fig. 19b) in order to keep the power output equal to the rated value. The same happens for the Mod-TI case. However, when $\gamma < -25°$, the turbine exits region III and enters region II. Therefore, as blade pitch (purple squares in Fig. 19b) reaches the value for maximum power coefficient, $\lambda$ starts decreasing (purple squares in Fig. 19a).

Figure 20 shows the results for the power and thrust loss factors in this scenario. Once again the proposed model exhibits a very good match with the experiments, falling within the uncertainty bands in most cases.

These results confirm the ability of the method to correctly represent the effects of different control approaches, covering both regions II and III, including derating. This is crucially important because, as shown, control laws have a strong impact on the behavior of power and trust in misaligned conditions.

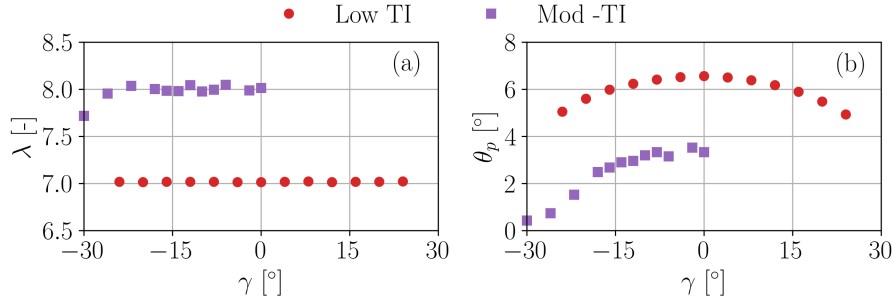

**Figure 19.** Experimental campaign 3. Blade pitch angle $\theta_p$ **(a)** and tip speed ratio $\lambda$ **(b)** in Low-TI and Mod-TI inflows.

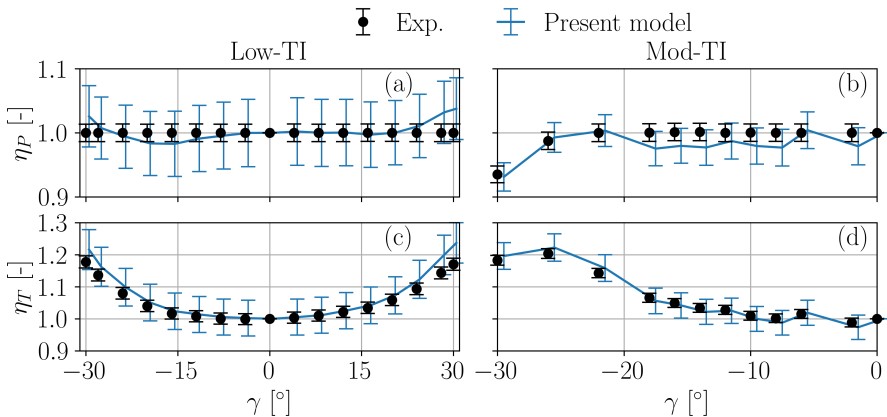

**Figure 20.** Experimental campaign 3. Power loss factor $\eta_P$ **(a-b)** and thrust loss factor $\eta_T$ **(c-d)** vs. yaw misalignment $\gamma$, in above-rated-speed operation for the Low-TI and Mod-TI inflow cases. Model predictions: solid lines; experimental measurements: black circles. Whiskers indicate the 95% confidence intervals.

## 5 Optimal wake steering

The insight provided by the new model suggests two questions:

– What is the power-optimal way to yaw a single turbine out of the wind?

– And does the new model affect the way wake steering should be conducted?

We try to give some initial answers to these questions in the following two sections.

### 5.1 Optimal power capture of a single misaligned turbine

The new model was used to compute the optimal power of a wind turbine when it is misaligned with respect to the wind. The analysis was conducted for the same IEA 3.4 MW wind turbine used for the previous numerical validation of the model.

The optimal control strategy was computed by numerical optimization using an adaptive Nelder-Mead algorithm (Gao and Han, 2012), and results are shown in Fig. 21. The figure reports also the standard region II control approach, which consists in holding the pitch angle fixed while the generator torque is varied proportionally to the square of the rotor speed, i.e. $Q \sim \Omega^2$, as explained in Sect. 3.3. Assuming as a first approximation that $P = \cos\gamma^{p_p}$, and considering that $Q = P/\Omega$, it follows that $\Omega \sim \cos\gamma^{p_p/3}$. The figure reports the solution computed for a coefficient $p_p = 1.88$, following Fleming et al. (2015).

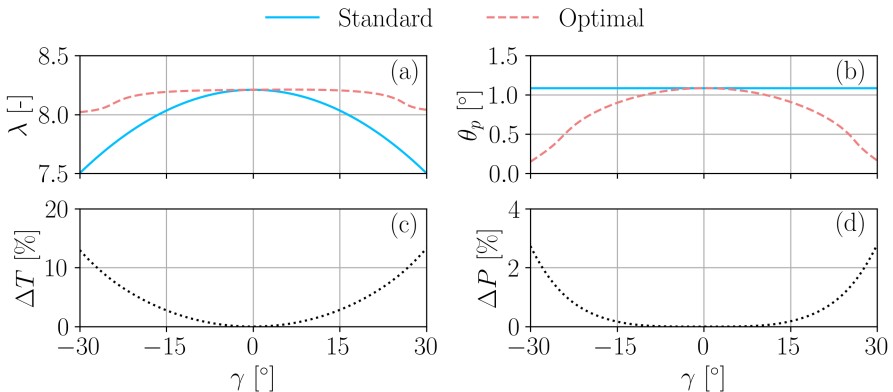

**Figure 21.** Comparison between standard and optimal control strategies for different yaw angles $\gamma$. Tip speed ratio $\lambda$ **(a)**; pitch angle $\theta_p$ **(b)**; percent thrust difference between the optimal and standard strategies **(c)**; percent power difference between the optimal and standard strategies **(d)**.

As the yaw misalignment increases, the tip speed ratio drops for the standard control strategy, driven by the reduced rotor-orthogonal component of the wind. Since the pitch angle remains fixed, the reduced $\lambda$ leads also to a decreased thrust coefficient. Although this might be beneficial for reducing loading on the yawed turbine, the resulting drop in power is significant. On the other hand, the optimal strategy governs the turbine to keep a much more constant tip speed ratio and thrust coefficient, while the blade pitches back a little. This results into some power boost, which is small for moderate angles but reaches above 3% around $\pm 30°$. These findings are in line with results presented by Cossu (2021a) and Heck et al. (2023). While the higher $C_T$ implies that the turbine is loaded more than in the standard case, it has also an effect on the wake that will be felt downstream, as explored in the next section.

## 5.2 Optimal power capture of two turbines

The previous section showed that a single turbine can extract more energy from the wind in misaligned conditions when its $C_T$ is increased compared to a standard region II control approach. This so-called overinductive yaw control (Cossu, 2021a) increases the velocity deficit in the wake, but it also affects its recovery and enhances its deflection. It is therefore necessary to find the optimal tradeoff among these complex effects when considering wake steering wind farm control (Meyers et al., 2022).

FLORIS v3 (NREL, 2023b), modified with the present model, was used to optimize the power capture of a cluster of two IEA 3.4 MW wind turbines placed at a distance of 5 diameters. The wake was modelled with the Gauss-Curl-Hybrid model (King et al., 2021). The inflow is characterized by an ambient wind speed $u_{\infty,\text{hub}} = 9.7\ \text{ms}^{-1}$, a shear of 0.12, and a turbulence intensity of 6%. A 60° range of wind directions $\Phi$ was considered, in order to realize different degrees of overlap between the wake and the downstream rotor. The optimal wind farm control strategy was computed by numerically maximizing the cluster power with the same adaptive Nelder-Mead algorithm used for the single-turbine case of the previous section.

Results are shown in Fig. 22. The plots report in green dotted lines the results obtained with greedy control (i.e., each turbine maximizes its own power capture), in blue solid lines the solution obtained with wake steering control based on the $\cos^{p_p}$ law using $p_p = 1.88$, and in red dashed lines with wake steering control based on the present model. For the three control strategies, results were validated with LES-ALM simulations run for five different wind directions, namely $\Phi = \{270 \pm 5.74, 270 \pm 2.5, 270\}°$, corresponding to rotor overlaps of 50%, 78.2%, and 100%, respectively. The LES-ALM results are indicated in the figure with markers, where the colors correspond to the control strategy.

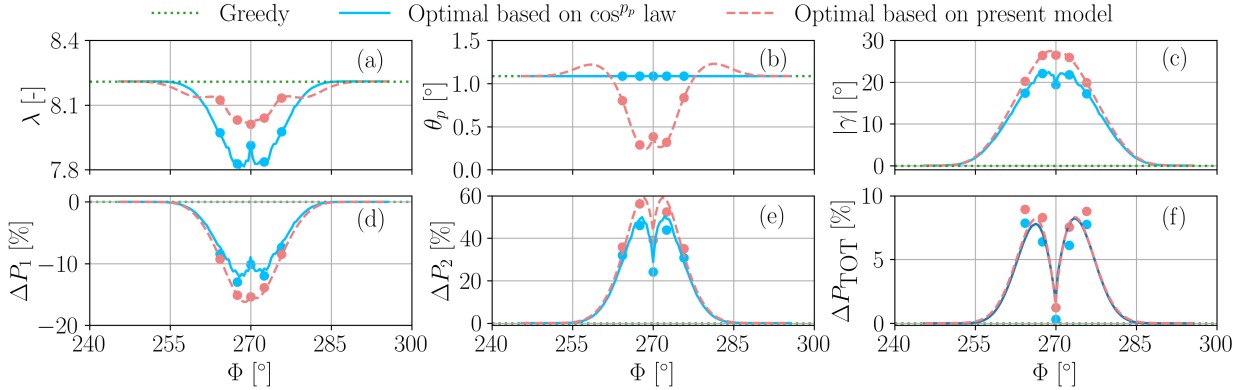

**Figure 22.** Control of a cluster of two turbines in wake interference conditions. Green: greedy policy; blue: optimal wake steering solution based on $\cos^{p_p}$; red: optimal wake steering solution based on the proposed model. Lines: FLORIS engineering wake model; markers: LES-ALM CFD. Tip speed ratio $\lambda$ (**a**); thrust coefficient $C_T$ (**b**); absolute yaw misalignment $|\gamma|$ (**c**); percent power changes with respect to the greedy policy for the upstream wake-steering turbine (**d**); percent power changes with respect to the greedy policy for the downstream turbine (**e**); overall percent power changes for the cluster of two turbines (**f**).

Figures 22a, 22b, and 22c respectively show the front turbine tip speed ratio $\lambda$, pitch angle $\theta_p$, and absolute misalignment angle $|\gamma|$, all plotted as functions of wind direction $\Phi$. The solution for the present model is characterized by a fairly constant tip speed ratio that, in conjunction with some pitch back of the blades at the highest misalignments, results also in a roughly constant thrust coefficient (not shown for brevity). This is in contrast with the solution based on the $\cos^{p_p}$ law, where both tip speed ratio and thrust coefficient drop at the higher misalignments that correspond to the strongest wake overlap conditions. In addition, the present model also results into slightly larger misalignment angles, as shown by Fig. 22c.

The bottom three plots show the effects of the various control strategies on power as function of wind direction $\Phi$. Figures 22d and 22e report the power changes with respect to the greedy strategy for the front and back turbines, respectively. It appears that the upstream machine, due to a larger misalignment, looses more power than in the $\cos^{p_p}$ case. Conversely, it also appears that the second machine gains more power with the strategy based on the new model, thanks to the larger misalignment of the upstream turbine but also due to its larger thrust coefficient. Finally, Fig. 22f shows the overall gain at the cluster level. Results indicate a fairly consistent improvement, in excess of roughly 1%, for almost the entire wake-overlap range.

The LES-ALM results confirm the findings based on the FLORIS engineering wake model: more power losses for the front turbine, more gains for the downstream one, resulting in a positive net gain for the cluster.

## 6 Conclusions

We have presented a new model to estimate the power performance of a misaligned wind turbine rotor. The model is a modified version of the classical blade element momentum theory, where the rotor is considered as a lifting wing of finite span operating at an angle of attack.

The new model reveals the following characteristics of the behavior of a misaligned rotor:

– Power does not depend on the misalignment angle according to the $\cos^{p_p}$ law, a formula in widespread use in the literature.

– The true effective misalignment angle that drives wake behavior is a combination of both yaw and tilt. Therefore, a two-dimensional wake model should be described in the plane formed by the rotor axis and wind vectors, not on the horizontal plane as commonly assumed.

– Power depends on the true misalignment angle, but – crucially – also on the way the rotor is governed as it is pointed out of the wind, a fact that probably explains the widely different performance observed by various authors. This fact also means that power losses due to misalignment can be mitigated by using a suitable control strategy.

– According to the model, the observed lack of symmetry between positive and negative misalignment angles is caused by the interaction with a sheared inflow. In these conditions, there is a complex interplay of various effects that may lead to different outcomes in terms of which yaw sign yields more or less power. In general, one can expect a small asymmetry at high thrust coefficients, while a more pronounced asymmetry emerges for low thrust and high tip speed ratios, where a higher power is generated for positive yaw angles. However, in general the behavior of power (but of thrust too, which also exhibits an asymmetric behavior) depends not only on the rotor design characteristics but also on the way it is governed, through the values of the pitch setting and of the tip speed ratio. Additionally, in the field other effects may be present (e.g., due to an asymmetric behavior of the onboard wind vane), which may add to the phenomena described by the model.

– A constant-over-the-rotor induction is sufficient to accurately describe the power and thrust behavior of a misaligned rotor in a sheared inflow. In fact, under classical small angle assumptions, the tilting of the inflow due to misalignment and shear has only a negligible effect on the quality of the results.

The model was derived in a semi-analytical form, leading to a closed system of equations that can be directly integrated with engineering wake models, at an irrelevant computational cost. To improve its accuracy, we proposed a specific implementation

that overcomes the intrinsic limits brought by the analytical solution of some model integrals. The proposed implementation corrects for the effects of misalignment a higher-fidelity power model obtained in aligned conditions, and calibrates the model parameters based on measurements.

The model was validated in a broad range of cases, considering LES-ALM numerical simulations of various multi-MW machines as well as experimental observations on a scaled wind tunnel model, in different inflows (from unsheared laminar to

715 sheared highly turbulent conditions), operating with controllers in the loop in regions II, III, as well as in derated conditions. In all cases, the model achieved a very satisfactory agreement with the numerical and experimental reference power and thrust values. Additionally, we demonstrated how the model can be integrated with given control laws, achieving an excellent match in the calculation of the setpoints. The model was also compared with a similar model recently developed by Heck et al. (2023), limitedly to the control region II where it is applicable, consistently improving on its predictions and exhibiting a wider

applicability to arbitrary control strategies.

Using the proposed model, we maximized the power capture of a wind turbine for a range of misalignment angles, obtaining the optimal power strategy in terms of pitch setting and tip speed ratio. Results indicate that the maximum power extraction is obtained by keeping an almost constant tip speed ratio and by slightly reducing the blade pitch as the turbine yaws out of the wind. This also implies a roughly constant thrust coefficient, which will increase the loading on the yawed turbine, but will

also have an effect on its wake.

Next, we applied the new model to the maximization of the power by wake steering for a cluster of two turbines. The resulting control strategy was compared to the one obtained by the classical $\cos^{p_p}$ power loss model, and validated by means of LES-ALM simulations for a few selected cases. Results indicate that the proposed model results in slightly greater power losses for the wake-steering turbine, which are more than compensated by greater power gains for the wake-affected one, achieving a

730 small but consistent gain in power at the cluster level for the full range of possible wake overlaps.

Future work should investigate the effects of the new model and its resulting control strategy in more complex conditions. Of particular interest is the analysis of the effects on loads, which might increase because of the eliminated drop in thrust coefficient as the turbine is yawed out of the wind.

## Appendix A: Transformation matrices

### A1 Transformation from ground to nacelle frame of reference

The nacelle-fixed frame of reference is obtained from the ground-fixed frame by a first rotation $\delta$ about the horizontal axis $\boldsymbol{y}_g$, followed by a rotation $\gamma$ about the vertical axis $\boldsymbol{z}_g$. The components of a generic vector $\boldsymbol{v}$ are noted $\underline{v}_g$ when measured in the ground frame $\mathcal{F}_g$, and $\underline{v}_n$ when measured in the nacelle frame $\mathcal{F}_n$. Combining the two successive rotations, one obtains the transformation of components from one frame to the other as

$$\underline{v}_n = \begin{bmatrix} \cos\delta & 0 & -\sin\delta \\ 0 & 1 & 0 \\ \sin\delta & 0 & \cos\delta \end{bmatrix} \begin{bmatrix} \cos\gamma & -\sin\gamma & 0 \\ \sin\gamma & \cos\gamma & 0 \\ 0 & 0 & 1 \end{bmatrix} \underline{v}_g = \begin{bmatrix} \cos\delta\cos\gamma & -\cos\delta\sin\gamma & -\sin\delta \\ \sin\gamma & \cos\gamma & 0 \\ \sin\delta\cos\gamma & -\sin\delta\sin\gamma & \cos\delta \end{bmatrix} \underline{v}_g. \tag{A1}$$

The inverse transformation is simply given by the matrix transpose.

Using Eq. (A1), the components of the ambient velocity vector $\boldsymbol{u}_\infty$ in the nacelle-attached frame are readily found to be $\underline{u}_{\infty n} = u_\infty \{\cos\delta\cos\gamma, \sin\delta, \sin\delta\cos\gamma\}^T$, where the scalar wind speed is $u_\infty = |\boldsymbol{u}_\infty|$, while $\underline{x}_{n_n} = \{1,0,0\}^T$. Hence, it follows that

$$\frac{\boldsymbol{u}_\infty}{u_\infty} \cdot \boldsymbol{x}_n = \cos\mu = \cos\delta\cos\gamma. \tag{A2}$$

### A2 Transformation from wake-deflection to ground frame of reference

The nacelle and wake-deflection frames share the same unit vector $\boldsymbol{x}_n = \boldsymbol{x}_d$, which corresponds to the rotor axis of rotation. The $\boldsymbol{z}_d$ unit vector is orthogonal to the plane composed by $\boldsymbol{u}_\infty$ and $\boldsymbol{x}_d$, and therefore it can be written as $\boldsymbol{z}_d = z\boldsymbol{x}_d \times \boldsymbol{u}_\infty$, where $z$ is a normalization scalar such that $\boldsymbol{z} \cdot \boldsymbol{z} = 1$. Performing the cross product and the normalization, one finds $\underline{z}_{d_n} =$

$z\{0, -\sin\delta\cos\gamma, \sin\gamma\}^T$ and $z = 1/\sqrt{\cos^2\gamma\sin^2\delta + \sin^2\gamma} = 1/\sin\mu$. A right-handed triad is completed by setting $\boldsymbol{y}_d = -\boldsymbol{x}_d \times \boldsymbol{z}_d$, which yields $\underline{y}_{d_n} = z\{0, \sin\gamma, \sin\delta\cos\gamma\}^T$. The transformation matrix between the wake-deflection and nacelle-fixed components is therefore readily obtained as

$$\underline{v}_n = \begin{bmatrix} 1 & 0 & 0 \\ 0 & z\sin\gamma & -z\sin\delta\cos\gamma \\ 0 & z\sin\delta\cos\gamma & z\sin\gamma \end{bmatrix} \underline{v}_d. \tag{A3}$$

Finally, the transformation between wake-deflection and ground-fixed components follows by using Eq. (A3) and Eq. (A1),
which yields

$$\underline{v}_g = \begin{bmatrix} \cos\delta\cos\gamma & \sin\gamma & \sin\delta\cos\gamma \\ -\cos\delta\sin\gamma & \cos\gamma & -\sin\delta\sin\gamma \\ -\sin\delta & 0 & \cos\delta \end{bmatrix} \begin{bmatrix} 1 & 0 & 0 \\ 0 & z\sin\gamma & -z\cos\gamma\sin\delta \\ 0 & z\cos\gamma\sin\delta & z\sin\gamma \end{bmatrix} \underline{v}_d, \tag{A4a}$$

$$= \begin{bmatrix} \cos\delta\cos\gamma & 1/z & 0 \\ -\cos\delta\sin\gamma & z\cos^2\delta\sin\gamma\cos\gamma & -z\sin\delta \\ -\sin\delta & z\sin\delta\cos\delta\cos\gamma & z\cos\delta\sin\gamma \end{bmatrix} \underline{v}_d. \tag{A4b}$$

The inverse transformation is simply given by the matrix transpose.

Using Eq. (A4b), the longitudinal (given by Eq. 11) and lateral (sidewash, given by Eq. 13) flow velocity components at the streamtube outlet can be transformed into the corresponding longitudinal and lateral components in the ground frame:

$$\frac{v_{o,g}}{u_\infty} = \frac{C_T}{4}\cos\delta\sin\gamma, \tag{A5a}$$

$$\frac{w_{o,g}}{u_\infty} = \frac{C_T}{4}\sin\delta. \tag{A5b}$$

## Appendix B: Effect of a non-uniform axial induction on the rotor disk

As mentioned in Sect. 2.4, misalignment and shear cause a non-uniform distribution of the induction over the rotor disk. Following the classical approach used for helicopter rotors in forward flight (Johnson, 1995), the simplest model of non-uniform axial induction is the one expressed by Eq. (6), based on the 1P harmonics $\kappa_{1s}$ and $\kappa_{1c}$. For the present application, however, it appears that the inclusion of these terms is not necessary. To show this, we consider the $\kappa_{1s}$ term, which is triggered by the misalignment $\mu$ and results in the largest induction in the most downstream portion of the rotor disk. Figure B1 presents the loss factors $\eta_T$ and $\eta_P$ predicted by the model with (red dotted line) and without (solid blue line) the 1P sine term $\kappa_{1s}$. Differences appear to be negligible, especially for $\eta_T$. Neglecting $\kappa_{1s}$ leads to a slight decrease in $\eta_P$ as the misalignment increases, reaching a maximum difference of 0.21% at $\gamma = -30°$.

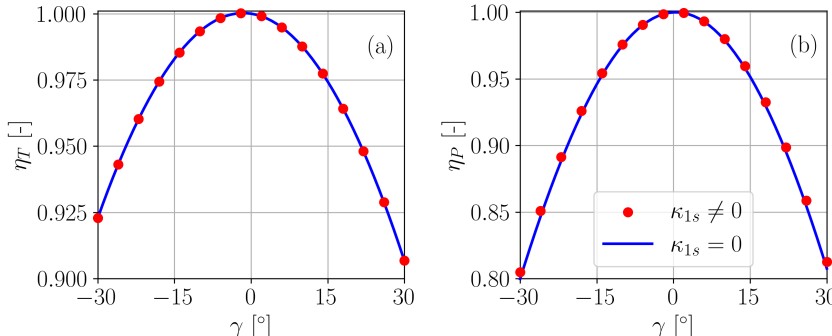

**Figure B1.** Thrust loss factor $\eta_T$ **(a)**, and power loss factor $\eta_P$ **(b)** computed with the proposed model, with and without the 1P sine harmonic $\kappa_{1,s}$ in Eq. (6). Results are computed for the IEA 3.4 MW reference wind turbine, subjected to a vertical shear $k = 0.2$, operating at a tip speed ratio $\lambda = 8.5$, and with a blade pitch angle $\theta = 1°$.

## Appendix C:  Experimental data set

Figures C1 and C2 report the blade pitch $\theta_p$ and tip speed ratio $\lambda$ measured in the experimental campaigns 1 and 2, respectively. The same quantities for experimental campaign 3 are reported in Fig. 19.

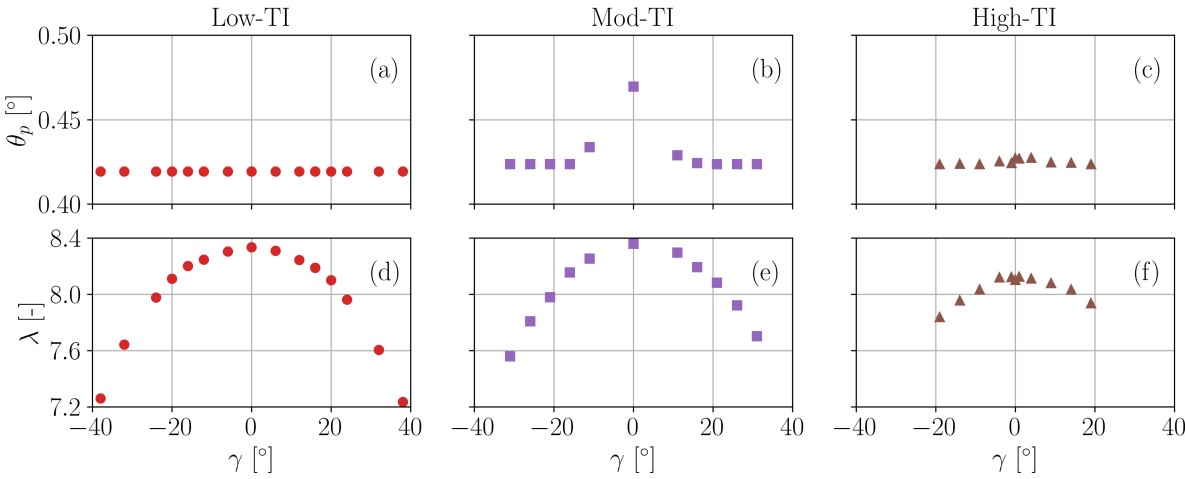

**Figure C1.** Experimental campaign 1. Blade pitch angle $\theta_p$ **(a)** and tip speed ratio $\lambda$ **(b)** in Low-TI, Mod-TI, and High-TI inflows.

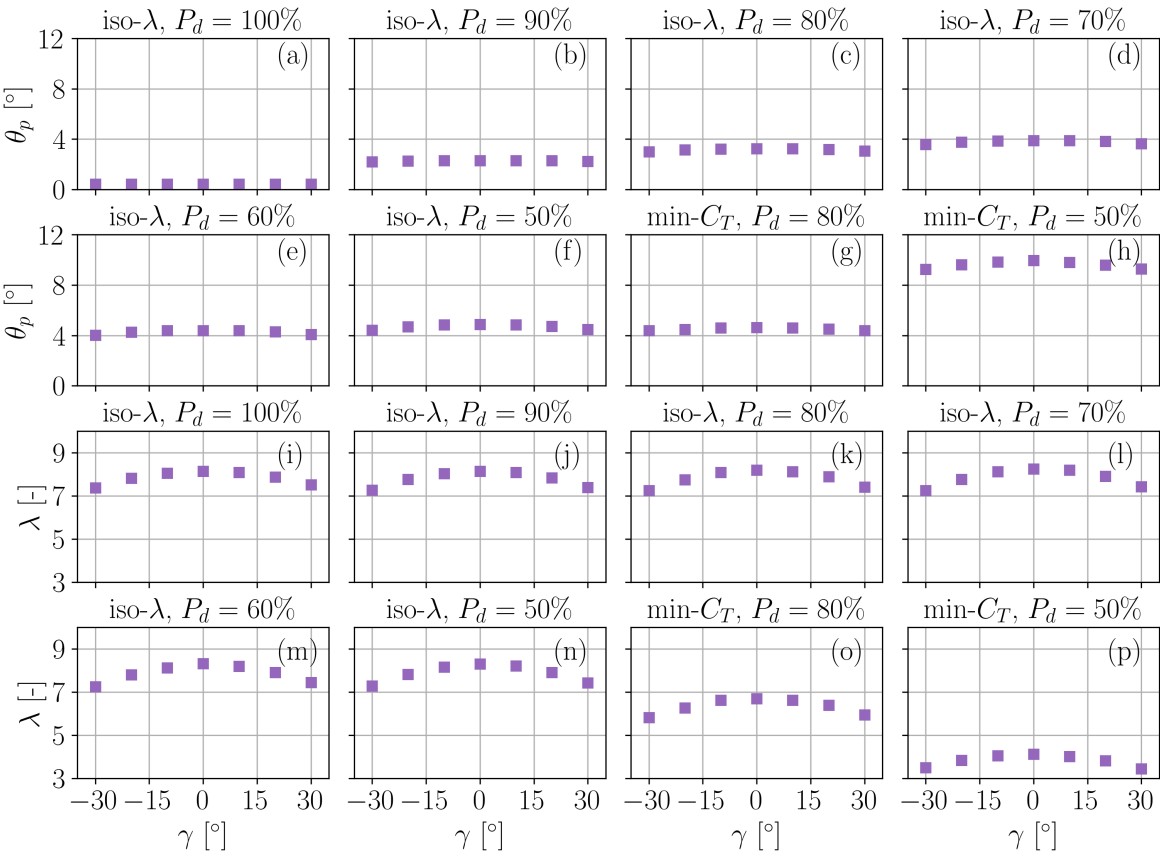

**Figure C2.** Experimental campaign 2. Blade pitch angle $\theta_p$ **(a)** and tip speed ratio $\lambda$ **(b)** in the Mod-TI inflow.

## Appendix D: Nomenclature

| | |
|---|---|
| $A$ | Rotor disk area |
| $a$ | Axial induction |
| $B$ | Number of blades |
| $C_D$ | Drag coefficient |
| $C_L$ | Lift coefficient |
| $C_{L,\alpha}$ | Lift slope |
| $C_P$ | Power coefficient |
| $C_T$ | Thrust coefficient |
| $C_T'$ | Modified thrust coefficient of Heck et al. (2023) |
| $D$ | Rotor diameter |
| $F_n$ | Normal force |
| $F_t$ | Tangential force |
| $K$ | Coefficient relating aerodynamic torque and squared rotor speed in control region II |
| $k$ | Linear vertical wind shear coefficient |
| $\dot{m}$ | Mass flux |
| $p$ | Pressure |
| $P$ | Power |
| $P_d$ | Power demand (derating) |
| $Q$ | Rotor torque |
| $R$ | Rotor radius |
| $r$ | Spanwise coordinate |
| $T$ | Thrust force |
| $u_\infty$ | Free-stream wind speed |
| $u_{\infty,\text{hub}}$ | Free-stream wind speed at hub height |
| $u$ | Longitudinal velocity component |
| $u_n$ | Rotor-orthogonal velocity component |
| $u_t$ | Rotor-tangential velocity component |
| $v$ | Lateral velocity component |
| $x$ | Cartesian coordinate |
| $y$ | Cartesian coordinate |
| $z$ | Cartesian coordinate |
| | |
| $\alpha$ | Angle of attack |

|  |  |  |
|---|---|---|
| | $\beta$ | Blade twist angle |
| 810 | $\delta$ | Rotor tilt angle |
| | $\eta_P$ | Power loss factor |
| | $\eta_T$ | Thrust loss factor |
| | $\gamma$ | Rotor yaw angle |
| | $\lambda$ | Tip speed ratio |
| 815 | $\mu$ | Rotor total (true) misalignment angle |
| | $\Omega$ | Rotor angular speed |
| | $\varphi$ | Inflow angle |
| | $\Phi$ | Wind direction |
| | $\psi$ | Rotor azimuth angle |
| 820 | $\rho$ | Air density |
| | $\theta$ | Local pitch angle |
| | $\theta_p$ | Blade pitch rotation at the pitch bearing |

|  |  |  |
|---|---|---|
| 825 | $\underline{v}_f$ | Components of vector $\boldsymbol{v}$ in frame $f$ |
| | $(\cdot)_d$ | Quantity evaluated in the wake-deflection intrinsic frame of reference |
| | $(\cdot)_g$ | Quantity evaluated in the ground frame of reference |
| | $(\cdot)_n$ | Quantity evaluated in the nacelle frame of reference |
| | $(\cdot)_{1c}$ | Once-per-revolution cosine harmonic |
| 830 | $(\cdot)_{1s}$ | Once-per-revolution sine harmonic |

|  |  |  |
|---|---|---|
| | 0P | Zeroth (constant) harmonic |
| | 1P | One per revolution harmonic |
| | ALM | Actuator Line Method |
| 835 | BEM | Blade Element Momentum |
| | CFD | Computational Fluid Dynamics |
| | CTA | Constant Temperature Anemometry |
| | FLORIS | FLOw Redirection and Induction in Steady State |
| | LES | Large Eddy Simulation |
| 840 | RPM | Revolutions Per Minute |
| | SOWFA | Simulator fOr Wind Farm Applications |
| | TI | Turbulence Intensity |

*Code and data availability.*  An implementation of the model described in this article in the FLORIS framework is available on Github at https://github.com/sTamaroTum/Beyond_the_cosine_law/ (Tamaro et al., 2024). The repository also contains all the data and the Jupyter notebooks used to generate the figures. The code and the scripts to reproduce the figures can be run on Binder at the link https://tinyurl.com/btcl-figs. The notebook of Fig. 5 can be used to interactively plot the thrust and power coefficients for other user-defined values of the model parameters, while the one of Fig. 6 can be similarly used to visualize different control trajectories. The notebooks of Figs. 21 and 22 contain also the code used for computing the optimal control policies. The complete data sets from the LES simulations and wind tunnel experiments are available upon request.

*Author contributions.*  CLB developed the formulation of the misalignment model, and supervised the overall research. ST implemented the model, performed the LES simulations and the corresponding model validation, and conducted the wake-steering analyses with FLORIS. FC performed the validation with respect to the experimental measurements. All authors contributed to the interpretation of the results. CLB and ST wrote the manuscript, with contributions by FC in the experimental section. All authors provided important input to this research work through discussions and feedback and by improving the manuscript.

*Competing interests.*  The authors declare that they have no conflict of interest, except for CLB who is the Editor in Chief of the Wind Energy Science journal.

*Financial support.*  This work has been supported in part by the PowerTracker project, which receives funding from the German Federal Ministry for Economic Affairs and Climate Action (FKZ: 03EE2036A). This work has also been partially supported by the MERIDIONAL project, which receives funding from the European Union's Horizon Europe Programme under the grant agreement No. 101084216.

*Acknowledgements.*  The authors express their gratitude to the Leibniz Supercomputing Centre (LRZ) for providing access and computing time on the SuperMUC Petascale System under Projekt-ID pr84be "Large-eddy Simulation for Wind Farm Control". The authors would like to thank K. Heck for the valuable discussion, and A. Guilloré and F. Mühle for the help in the measurement campaign.

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
