# Peer review of "On the power and control of a misaligned rotor – Beyond the cosine law –"

_Wind Energy Science, 2023_

## Referee Comment (RC2)

Title: On the power and control of a misaligned rotor - Beyond the cosine law
Author(s): Simone Tamaro et al.
MS No.: wes-2023-133
MS type: Research article

The submitted paper studies the power of wind turbines in yaw misalignment. An induction model is coupled with a simple blade element model, and the resulting model outputs are compared to large eddy simulations (LES) of actuator line modeled (ALM) turbines and wind tunnel experiments. Overall, the paper could be useful for the community. Modelling the power of yawed turbines is important. The contribution of this paper is coupling an induction model with a blade element model. The LES and experimental validation campaign is thorough.

However, it appears that several components of the proposed model have been already developed in the following two papers:

1. Shapiro, Gayme & Meneveau *Journal of Fluid Mechanics* (2018) [1]
   This paper develops a lifting line model for the transverse velocity (downwash) associated with a yawed turbine. The present paper appears to follow the same analysis, resulting in the same final answer (compare: Eq. 2.7 in Shapiro to Eq. (13) here), but reference to Shapiro *et al.* (2018) [1] is missing.

2. Heck, Johlas & Howland *Journal of Fluid Mechanics* (2023) [2]
   This paper develops a model for the induction, thrust, power, and wake velocities for a yawed actuator disk, using the lifting line model of Shapiro, Gayme & Meneveau (2018) [1], but also accounting for how the induction affects the transverse velocity. The induction and wake velocity model developed in Heck, Johlas & Howland (2023) [2] is the same as the induction and wake velocity model in this present paper (compare: combining Eqs. (2.15) and E1 in Heck to Eq. (14) here), but this is not stated in the current manuscript.

To summarize, Section 2.5 in the present paper can be replaced with references to [1] and [2]. Explaining the progress of yaw modelling based on existing literature and how the present paper has contributed will be helpful for the readers. Overall, the main contribution of this submitted paper is to build on Shapiro, Gayme & Meneveau (2018) [1] and Heck, Johlas & Howland (2023) [2], by coupling their induction model with a simple blade element model, and the detailed comparisons to LES and experimental data. These are useful contributions to the literature, but the framing, comparisons to baseline methods, and other comments below should be re-considered in the authors' revision.

**General comments**

1. The authors have not compared their modified model (coupling with a simple blade element model) version to baseline approaches, including the Glauert induction model (model for rotor averaged induction in yaw) and the actuator disk induction model from Heck, Johlas & Howland

(2023) [2] (same as present model without the blade element coupling).

2.  It appears that tunable parameters in the model are calibrated based on the same data that they are tested against, which is not ideal practice. Can this be considered model validation? How should this be done in general?

3.  Many figures are quite small, making it challenging to discern the accuracy of the model.

4.  The paper states that a major contribution of the modelling is to capture asymmetry from wind speed shear, but the quantitative effect of wind speed shear presented in the results seems to be (visually) very small. I suggest quantifying its impact to help see its effect more clearly.

**Line comments**

1.  Line 28: The formatting of the power equation looks as though all the variables are in the denominator.

2.  Line 73: *"Additionally, the new model clarifies the behavior of power capture with respect to some rotor design parameters and – even more importantly – with respect to the way a rotor is governed when it is misaligned. This is an effect that has been neglected in all analyses conducted so far, and that most probably explains the large scatter observed by various author"*

    This statement is not correct. Howland *et al.* (2020) [3] developed a blade element model for the power-yaw relationship that incorporates rotor design parameters and the way a rotor is governed in misalignment. However, Howland *et al.* (2020) [3] did not have a model for induction in yaw.

3.  Section 2.2: Why have the authors assumed inflow with linear vertical shear? This seems to be a limiting decision in the context of a paper which focuses on building a model for yawed turbines in general. This should be justified in more detail. For example, Liew *et al.* (2020) [4] identified that waked inflow modifies the power-yaw relationship, but this inherently cannot be captured in the present model that only considers linear vertical shear. Also, wind shear in the stratified boundary layer is very rarely linear.  In the modelling and field experiment study of Howland *et al.* (2020) [3], the joint effects of low-level jets and wind veer were found to be important. They can be modeled using blade element modelling [3].

4.  Section 2.2: Why have the authors elected to neglect wind veer, which has been shown to be important in wake steering [5] and in power-yaw modelling [3] in published papers?

5. Equations 5a and 5b: Tangential induction has been neglected. This should be mentioned and justified.

6. Section 2.4: The structure of presentation in this section is a little odd. It starts by claiming that the non-uniform induction does not affect the results, then shows the equations, then neglects it for the remainder of the study. I suggest moving this section to the Appendix, and also including the quantitative evidence (referenced but not shown) that it is negligible in your cases.

7. Page 9 footnote: *"This interpretation also reveals that the so-called curled shape of laterally deflected wakes (see e.g. Martínez-Tossas et al. (2021) and references therein) is nothing else than the effect of the horseshoe vortex structure generated behind a lifting wing, albeit with the addition of the swirl caused by the rotor rotation."*
This explanation is exactly the one provided by Shapiro *et al.* (2018) [1] which the authors have not referenced in their study. This statement should be removed and references to Shapiro *et al.* (2018) [1] must be added.

8. Equation (13): This is the same lateral velocity equation derived by Shapiro *et al.* (2018) [1] derived using Prandtl's lifting line theory, except that Shapiro *et al.* (2018) [1] assumed that thrust varies with cos(yaw)^2. Heck *et al.* (2023) [2] extended the lifting line model of Shapiro *et al.* (2018) [1] to no longer assume thrust follows cos(yaw)^2, and the final answer in Heck *et al.* (2023) [2] (Eq. (2.15)) is the same as Eq. (13) here.

9. Equation (14): Similarly, this is the same induction model derived by Heck *et al.* (2023) [2], although it is presented in a slightly different form in the previously published paper. Appendix E from Heck *et al.* (2023) [2] is pertinent (i.e. combine Eq. (2.15) with Eq. (E1) to arrive at the induction model form below that can be compared to Eq. (14) in the present paper).
From Heck *et al.* (2023) [2], the induction model equation is:

$$a_n = \frac{2C_T - 4 + \sqrt{16 - 16C_T - C_T^2 \sin^2(\gamma)}}{-4 + \sqrt{16 - 16C_T - C_T^2 \sin^2(\gamma)}}$$

which appears to yield identical predictions to Equation (14) in this study.
In summary, Section 2.5 is a repeat of existing literature and can be removed, with appropriate references added.

10. Line 206: Please justify the assumption of the small inflow angle, especially in the context of yaw and tilt misalignment and shear.

11. Equations (18a) and (18b): Have the authors assumed that the lift and drag coefficients are constant along the wind turbine blade? Please explain.

12. Line 239: *"The power model reveals that vertical shear is the culprit for the observed lack of symmetry with respect to yaw misalignment."*

I don't quite understand this sentence. When the authors state " observed lack of symmetry," are they referring to existing published literature or to their own data (which to this point has not been presented). Previous studies have already explained and modeled that wind speed shear and wind direction veer cause the asymmetric power with respect to yaw misalignment. This current study neglects veer, which also seems limiting.

13. Figure 5: The effect of wind speed shear is very small. I expect the effect of veer is much larger, and especially when there is both shear and veer.

14. Paragraph beginning on line 274: It's good the authors state assumptions and simplifications here, but they should also all be stated and explained within the derivation. Otherwise, it seems as if the authors are making ad hoc choices about what is important and what approximations are made.

15. Equation (28) and associated discussion: I do not understand the motivation for simplifications to be applied to the model and then the tuning of more unknown parameters. How does this affect the result? How can this be done in general? Do the authors expect these parameters to be universal, and if not, how can this model be applied to a new wind turbine model? Do we need power data for turbines in yaw misalignment to tune this model? If so, that is not necessarily useful as a predictive model.

16. The authors need to include an Appendix that describes the tuning process in much more detail. What do the authors mean when they say *"a different random 50% subset of the available data"*? Is this the training-testing split? Presumably the authors are not performing model tuning with the same data that are used to test the model accuracy, as this is improper practice and can bias the results. In the added Appendix I am requesting, model results without any tuning must be shown.

17. $\beta$: It strikes me as a bit strange to have a tunable parameter in the model represent a known geometrical feature such as blade twist.

18. Figure 7: the authors show four results in this figure that are almost identical. It is very challenging to discern any notable differences among the subfigures, so it's reasonable to ask whether the authors have really tested the limits and applicability of their modelling framework. For example, why has the tip-speed ratio been kept within such a small region?

19. Line 324: *"However, as thrust is decreased (and pitch increased), power capture at positive $\gamma$ values is larger than for negative misalignments."*
The asymmetry is almost not visible in Figure 7 to me. Perhaps quantify to make it more clear?

20. Line 327: "trust coefficient" -> "thrust coefficient"

21. Figure 8: Why have the authors chosen to only consider low thrust coefficients with a maximum of C_T=0.6? It is interesting to show higher thrust coefficients. For example, Heck *et al.* (2023) [2] found that the thrust should increase with yaw to reduce the power loss. However, it seems that their induction model (same as your Equation (14)) is less accurate at higher C_T.

22. Line 341: *"[...] and is capable of describing even relatively minor effects of the complex behavior of a misaligned wind turbine rotor in a sheared inflow."*
I did not follow what the authors meant by this statement

23. Line 345: Since the rotor aerodynamic characteristics are necessary in your model, please include an Appendix which describes all relevant characteristics in this paper, so that the paper is self-contained.

24. What is the thrust coefficient of the experimental turbine?

25. Figure 11: It seems that a lot of tunable parameters are fit within this model. I am again unclear as to what is within sample of tuning and what should be considered as model validation (which requires out of sample data).

26. Figure 13: The results are summarized as having "very good" agreement with limited discussion, but there are several occasions where the model predictions are outside experimental uncertainty. It would be better to discuss these in detail.

27. Section 4.1: The results of this section align exactly with the published study of Heck *et al.* (2023) [2], who found that the thrust coefficient should be increased as the yaw is increased to reduce the power loss.

28. Section 4.2: How does the operation of the leading turbine affect the wake? The specific equations should be shown. It should affect the initial streamwise and spanwise wake deficits. This was done in Heck *et al.* (2023) [2] (Figure 9 and Appendix C).

29. Figure 15: It seems that there is almost no benefit from the modified model compared to baseline FLORIS, from the lines in Figure 15(f). Why is the additional benefit so negligible?

30. Line 444: *"The LES-ALM results confirm the findings based on the FLORIS engineering wake model: less power losses for the front turbine, and more gains for the downstream one."*
Looking at Figure 15, this statement appears to be incorrect. It seems that the "Opt. (Model)" approach actually *increases* the turbine 1 power loss and increases turbine 2 power gain. Rather than decreasing power loss for the front turbine with more gains for the downstream one.

31. References: I am not sure why the authors have chosen to cite *arXiv* versions of papers that have been published before this present paper was submitted, but that must be corrected.

32. I recommend a title change, since other published papers have previously gone 'beyond the cosine law.' It is better to be specific about what contributions this paper contains.

**References**

[1] Shapiro, Carl R., Dennice F. Gayme, and Charles Meneveau. "Modelling yawed wind turbine wakes: a lifting line approach." *Journal of Fluid Mechanics* 841 (2018): R1.

[2] Heck, Kirby S., Hannah M. Johlas, and Michael F. Howland. "Modelling the induction, thrust and power of a yaw-misaligned actuator disk." *Journal of Fluid Mechanics* 959 (2023): A9.

[3] Howland, Michael F., Carlos Moral González, Juan José Pena Martínez, Jesús Bas Quesada, Felipe Palou Larranaga, Neeraj K. Yadav, Jasvipul S. Chawla, and John O. Dabiri. "Influence of atmospheric conditions on the power production of utility-scale wind turbines in yaw misalignment." *Journal of Renewable and Sustainable Energy* 12, no. 6 (2020).

[4] Liew, Jaime, Albert M. Urbán, and Søren Juhl Andersen. "Analytical model for the power–yaw sensitivity of wind turbines operating in full wake." *Wind Energy Science* 5, no. 1 (2020): 427-437.

[5] Archer, Cristina L., and Ahmad Vasel-Be-Hagh. "Wake steering via yaw control in multi-turbine wind farms: Recommendations based on large-eddy simulation." *Sustainable Energy Technologies and Assessments* 33 (2019): 34-43.

---

## Author Comment (AC1)

**Reply to the reviewers' comments for paper "On the power and control of a misaligned rotor – Beyond the cosine law"**

The authors would like to thank the two reviewers for their time and for the useful feedback. All inputs that they provided have contributed to the improvement of the paper.

A list of point-by-point replies to the reviewers' comments is reported in the following. The reviewer's comments are in black, and our replies in blue.

In addition to the extensive rewriting of entire sections, which also contain new results and additional bibliographical references, we have taken the opportunity of this deep revision to make several small editorial changes to the text in order to improve readability.

A revised version of the manuscript is attached to the present reply, with additions highlighted in blue and deletions marked in red.

The authors

Reviewer 1

The paper presents a new analytical relationship to predict the variation of thrust and power coefficients with the yaw angle. This is an important and timely topic, as an accurate estimation of these two dimensionless quantities is of great importance in implementing wake steering. The power coefficient directly determines the amount of power loss on the turbine that is yawed, and the thrust coefficient is an important parameter in engineering wake models used to estimate the amount of wake deflection in yawed conditions. Two assumptions are made to develop the model: (i) the relative flow angle is small, which is valid for high-tip speed ratios, and (ii) the induction factor is uniformly distributed across the rotor disk. The latter assumption is supported by a discussion on the significance of the non-uniformity for misaligned rotors. Model predictions are compared with both LES and wind tunnel data. Overall, the paper is well-written. The work is timely and of interest to the wind energy community. An interesting coordinate transformation was performed to group both yaw and tilt effects into one combined angle (called the misalignment angle) in a bespoke coordinate system. It is great to see that all coordinate systems and parameters are very carefully and clearly defined. Detailed discussions were made on the impact of inflow shear and its effect on the asymmetrical distribution of thrust and power. I believe this work will have a high impact, especially in the field of wake steering. However, before publication, I would like to ask the authors to address the following comments (sorted by the line number) to improve the clarity and completeness of their work.

Thank you for the useful feedback.

1. Line 44: The paragraph is too short. It can be merged with the following one.
   We have merged the paragraphs accordingly.

2. Line 84: Verb missing in "... and Sect. 3 for its validation..."
   We have edited the text accordingly.

3. Line 112 and 117: Line 112 says that Mu is always positive, but in line 117, it has a negative value.

*True, this is a mistake, we have edited the text accordingly.*

4. Line 116 and Figure 2: I agree with the argument that different distributions of yaw and tilt angles produce a similar wake in the II plane if their misalignment angle is the same. However, this is only valid if we neglect ground effect. Cases with a large tilt angle may have a strong interaction with the ground. This can be clarified here.
   *Thank you, we agree and we have edited the text accordingly.*

5. Line 128: The radial and azimuthal coordinates can be shown in Figure 1d.
   *We have modified the figure and the text as requested.*

6. Line 125: A brief discussion on the relevance of linear shear in comparison to widely used power law or logarithmic profiles can be mentioned here. In other words, you can better justify why linear shear was used instead of those profiles.
   *The choice was simply based on a desire to reduce the complexity of the derivations, and this has now been noted in the text.*

7. Line 150: "r" is first defined in 128 and later in 150 with apparently two different definitions.
   *In both cases $r$ represents the radial coordinate of the rotor plane. We have now removed the second definition of $r$, which was unnecessary and confusing.*

8. Line 199: I am not sure if I understand what 0P means here. Some clarification in the text would be useful.
   *As written, 0P (or zero P) stands for "constant-over-the-rotor". This term is also defined in the "Nomenclature" section.*

9. Figure 4: I think the angle shown in the figure should be mu instead of gamma.
   *Thank you, we have corrected the figure.*

10. Line 207: Authors can better justify the validity of the small inflow angle approximation. For instance, the value of phi for the blade tip of a turbine with a TSR of 8 can be reported here.
    *Thank you for the suggestion, we have added a quantification of the angle for a typical operating condition.*

11. Line 210: Typo: "we" and "list" should be replaced with "be" and "lift".
    *Thank you, we have corrected the typos.*

12. Line 210: I believe the assumption of C_L=C_L,alpha*alpha is only valid for a symmetrical aerofoil. This can be clarified in the text.
    *We actually measure the angle of attack with respect to the zero-lift direction. In this case, the expression C_L=C_L,alpha*alpha is valid also for a non-symmetric airfoils. This choice was made to eliminate one extra term from the (already quite complex) expressions. We understand however that we initially did not clearly explain this fact, so we have now expanded the text accordingly.*

13. Effect of tilt angle: It is good to consider the uptilt angle effect, but it would be useful to show how big the effect is on C_T and C_P. Arguably, the effect on turbine performance and its wake should not be very significant. With that in mind, I suggest mentioning that the developed model will also be very useful for tilt angle control, which could become popular in the future generation of wind turbines, especially in floating turbines.

Thank you for the suggestion, we have expanded the text as per your recommendation.

14. Line 315: It is useful to report the difference between the values of C_D and C_L,alpha obtained from the optimization study with the typical values reported for the blade aerofoil in 2D studies.
We have now greatly expanded this part, as part of a complete writing of the previous Sect. 2.9, now expanded into the new Sect. 3. The new sections 3.4 and 3.5 explain both a data-driven calibration of the model parameters and a simplified choice of the same parameters, the latter designed to ease the use of the model when tuning data is not available. In the new Sect. 4.2 we now provide a comparison between calibrated and actual 2D values, and we demonstrate the performance of the simplified choice of parameters by comparison with LES simulations. Thank you for this suggestion, because we believe that the explanation of the choice of model parameters is of great importance for the practical use of the proposed model.

15. Section 4.1: The discussion in this section is interesting. One thing that I however struggled to understand (perhaps I'm missing something here) is that Figure 15d shows that the power loss for the first turbine is more significant when the optimal model is used. However, this contrasts with what is mentioned in lines 439 and 488.
Thank you for spotting this mistake, the text has now been corrected.

16. Section 4.1 C_T effect: The manuscript points out that the thrust coefficient for the optimal model used in section 4.1 is higher for yawed cases compared with the standard approach. It is mentioned that C_T has an effect on the wake felt downstream. I agree with the authors, but I suggest elaborating on this more. As we know, C_T has two effects on the wake: (i) it increases the streamwise velocity deficit and (ii) also increases the amount of wake deflection. These two have counteracting effects on the downwind turbine. It is interesting to understand this effect in greater detail, at least discussing it in the paper.
Thank you for the suggestion, we have expended the text as suggested.

17. Missing references: I appreciate that the authors did a thorough literature review, but some relevant references are missing. For instance, the streamtube model in section 2.5 and finding the wake spanwise velocity in the farm wake based on lifting theory are highly relevant to Shapiro et al. (2018). For instance, eq 14 in this manuscript could be compared with equation 2.13 in that paper. Heck et al. (2022): It is true that your developed model improving the state-of-the-art by including tilt and shear effects in greater details. However, I believe it is still informative for the reader to show whether your model, compared with the one proposed in Heck et al., provides similar predictions if they are used for similar operating conditions.
Thank you for these relevant comments. We have now deeply rewritten the previous Sect. 3. The new Sect. 3 provides a much more in depth description of the implementation of the model, and includes an extensive discussion on the integration with arbitrary control strategies (now in Sect. 3.2). This has also the purpose of clarifying one of the main differences between the proposed model and the one of Heck et al. 2023, which was not explained clearly enough in the previous version of the manuscript. Specifically, the method of Heck et al. 2023 is limited to operation in region II, when the modified trust coefficient that is at the basis of their formulation remains constant as a function of the misalignment angle.

On the other hand, our formulation includes a complete thrust model, and takes as input the tip speed ratio and pitch angle. This crucial difference means that our method can be used with any desired control strategy, in region II, III, II1/2 (for noise mitigation, thrust clipping, etc.), and also in derated conditions. These important differences between the two methods are now briefly mentioned in the introduction and conclusions, and explained in great detail in Sect. 3.2. The new Sect. 4 presents detailed comparisons between our model and the one of Heck et al. 2023, whenever the latter is applicable. We believe that these extensive modifications have greatly improved the paper.

Reviewer 2
The submitted paper studies the power of wind turbines in yaw misalignment. An induction model is coupled with a simple blade element model, and the resulting model outputs are compared to large eddy simulations (LES) of actuator line modeled (ALM) turbines and wind tunnel experiments. Overall, the paper could be useful for the community. Modelling the power of yawed turbines is important. The contribution of this paper is coupling an induction model with a blade element model. The LES and experimental validation campaign is thorough.

However, it appears that several components of the proposed model have been already developed in the following two papers:
1. Shapiro, Gayme & Meneveau Journal of Fluid Mechanics (2018) [1]
This paper develops a lifting line model for the transverse velocity (downwash) associated with a yawed turbine. The present paper appears to follow the same analysis, resulting in the same final answer (compare: Eq. 2.7 in Shapiro to Eq. (13) here), but reference to Shapiro et al. (2018) [1] is missing.
2. Heck, Johlas & Howland Journal of Fluid Mechanics (2023) [2] This paper develops a model for the induction, thrust, power, and wake velocities for a yawed actuator disk, using the lifting line model of Shapiro, Gayme & Meneveau (2018) [1], but also accounting for how the induction affects the transverse velocity. The induction and wake velocity model developed in Heck, Johlas & Howland (2023) [2] is the same as the induction and wake velocity model in this present paper (compare: combining Eqs. (2.15) and E1 in Heck to Eq.(14) here), but this is not stated in the current manuscript.

To summarize, Section 2.5 in the present paper can be replaced with references to [1] and [2]. Explaining the progress of yaw modelling based on existing literature and how the present paper has contributed will be helpful for the readers. Overall, the main contribution of this submitted paper is to build on Shapiro, Gayme & Meneveau (2018) [1] and Heck, Johlas & Howland (2023) [2], by coupling their induction model with a simple blade element model, and the detailed comparisons to LES and experimental data. These are useful contributions to the literature, but the framing, comparisons to baseline methods, and other comments below should be re-considered in the authors' revision.

Thank you for your comments. As explained when answering to reviewer 1, we have now deeply rewritten extensive parts of the paper. We believe that our deep revision clarifies the differences between previous publications and our proposed method. The extensive comparisons with the work of Heck et al. 2023 also provides ample evidence of the much broader applicability and superior accuracy of our approach.

In passing, we would also like to mention that the score of "poor" given by this reviewer seems not to be objective nor fair. Although the previous version of the manuscript certainly had deficiencies, this reviewer should have recognized that this paper improves on the state of the art. We hope that this extensively revised version makes this point even clearer.

**General comments**

1.  The authors have not compared their modified model (coupling with a simple blade element model) version to baseline approaches, including the Glauert induction model (model for rotor averaged induction in yaw) and the actuator disk induction model from Heck, Johlas & Howland (2023) [2] (same as present model without the blade element coupling).

    Thank you for the comment. Unfortunately, in the previous version of the manuscript we gave for granted that a reader would be able to appreciate the main difference between our approach and the one of Heck et al. 2023. This was a mistake, for which we apologize. We have now tried to correct this error by expanding the text in various parts (see especially the new Sect. 3.2), and by adding extensive comparisons with Heck et al. 2023 in the new Sect. 4. In reality, also the reviewer seems to have missed the main point. In fact, the statement "same as present model without the blade element coupling" (which seems to imply a minor difference between what we proposed and what was written by Heck et al.) fails to consider the following:

    a.  Not having a blade element model (which our approach does have), means that Heck et al. 2023 is forced to assume that their modified thrust coefficient CT' remains constant as the turbine yaws out of the wind. Unfortunately, this is hardly true in general, and only applies (under some conditions) when the turbine operates in region II.

    b.  Crucially, our proposed approach does include a blade element model, which brings into the formulation the dependency on the pitch setting and tip speed ratio. This means that our method, contrary to Heck et al 2023, is applicable to any operational region, and with arbitrary control strategies in those regions.

    We believe that the new writing finally clarifies the fundamental points listed above. These theoretical differences between the two methods are also supported by new extensive comparisons among LES simulations, wind tunnel experiments, our model and the one of Heck et al 2023, in the cases where the latter is applicable.

    In addition, we also note that, as explained in the new Sect. 4.3, our method is also capable of excellent performance in the extreme case of a large dependency of the aerodynamic characteristics on the Reynolds number. Although this is a rather specific case, which is limited to small-scale wind tunnel models, it is again only due to the presence of a blade element model in our approach. On the other hand, the use of the global CT' parameter used by Heck et al. 2023, is blind to such effects, because it does not include the blade aerodynamic characteristics. Therefore, even from this point of view, "same as present model without the blade element coupling" is not a fair assessment of the crucial differences between the two approaches.

    Regarding the comparison with Glauert, we believe that the method of Heck et al. 2023 is much more sophisticated than Glauert, and adding other lines to the plots would decrease their readability, without adding any useful information.

2.  It appears that tunable parameters in the model are calibrated based on the same data that they are tested against, which is not ideal practice. Can this be considered model validation? How should this be done in general?

    Thank you for this comment. We have added Sect. 3.3 and 3.4, which expanded on the problem of model calibration. In addition, the new Sect. 4 includes new examples where the model is compared to data that was not used for calibration.

3.  Many figures are quite small, making it challenging to discern the accuracy of the model.

    Thank you for the comment. We have regenerated most of the plots and provided new ones, also enlarging them when necessary. We hope the new version is now more readable.

4. The paper states that a major contribution of the modelling is to capture asymmetry from wind speed shear, but the quantitative effect of wind speed shear presented in the results seems to be (visually) very small. I suggest quantifying its impact to help see its effect more clearly.

Thank you for this comment. As replied in response to point 1 above, in reality the main contribution of the method is probably not the one regarding shear, but the very broad applicability to any operating conditions and control strategy. In any case, we have now added the new Figs. 10 and 18, where we more clearly show the effects of shear.

**Line comments**

1. Line 28: The formatting of the power equation looks as though all the variables are in the denominator.

We have modified the text accordingly.

2. Line 73: "Additionally, the new model clarifies the behavior of power capture with respect to some rotor design parameters and – even more importantly – with respect to the way a rotor is governed when it is misaligned. This is an effect that has been neglected in all analyses conducted so far, and that most probably explains the large scatter observed by various author" This statement is not correct. Howland et al. (2020) [3] developed a blade element model for the power-yaw relationship that incorporates rotor design parameters and the way a rotor is governed in misalignment. However, Howland et al. (2020) [3] did not have a model for induction in yaw.

Thank you for the comment. The difference with respect to Heck et al. 2023 has now been explained in detail (see also our reply to point 1 above).

3. Section 2.2: Why have the authors assumed inflow with linear vertical shear? This seems to be a limiting decision in the context of a paper which focuses on building a model for yawed turbines in general. This should be justified in more detail. For example, Liew et al. (2020) [4] identified that waked inflow modifies the power-yaw relationship, but this inherently cannot be captured in the present model that only considers linear vertical shear. Also, wind shear in the stratified boundary layer is very rarely linear. In the modelling and field experiment study of Howland et al. (2020) [3], the joint effects of low-level jets and wind veer were found to be important. They can be modeled using blade element modelling [3].

Thank you for the comment. We have opted for a linear shear distribution in order to simplify the equations of the model. In fact, the coordinate transformation to move from the ground-fixed reference frame to the wake-deflection intrinsic frame introduces cumbersome equations. We have also used a power-law in Eq. (2), observing similar results to the linear approximation, but with much heavier and complex final equations. For these reasons, we believe the present approximation to be appropriate, as it favors the understanding and usability of the model. The inclusion of low-level jets and veer is also clearly possible, once again at the price of much more involved equations.

We thank the reviewer for the suggestion of expanding the work in this direction, which is certainly relevant and might extend the applicability of our approach. However, we consider these aspects -that would not impact the main assumptions and theory of the method- as out of scope here, and material for future work.

4. Section 2.2: Why have the authors elected to neglect wind veer, which has been shown to be important in wake steering [5] and in power-yaw modelling [3] in published papers?

Please see the reply to the previous point.

5. Equations 5a and 5b: Tangential induction has been neglected. This should be mentioned and justified.

A discussion on the neglection of tangential induction, together with various other effects, has been added at the end of Sect. 2.4. A similar discussion is repeated in the new Sect. 3.2, when explaining the use of loss functions. In a nutshell, it is clear that tangential induction, together with a myriad other effects, could be added to the present formulation. However, this way the model would be become a fully-blown BEM implementation, together with its complexity and computational cost. Notwithstanding these limitations, the results indicate that the present simplified analytical model is able to capture the trends of thrust and power changes remarkably well, considering yaw misalignment, arbitrary control setpoints, and shear.

6. Section 2.4: The structure of presentation in this section is a little odd. It starts by claiming that the non-uniform induction does not affect the results, then shows the equations, then neglects it for the remainder of the study. I suggest moving this section to the Appendix, and also including the quantitative evidence (referenced but not shown) that it is negligible in your cases.

We respectfully disagree on the comment on the structure of the presentation. We start with a non-uniform formulation of the induction, because this is what one would expect based on the theory of helicopters (for which, since they typically fly in wind-misaligned conditions, these similar analyses are text-book material). However, we have verified that the addition of the 1P harmonics is not strictly necessary in the present context, and we have explained this finding in Sect. 2.4, thereby dropping these extra terms from the subsequent derivations. Therefore we believe that the present explanation is logical, and provides the necessary context to the reader.

However we agree with the other suggestion by the reviewer, and we have now added a new appendix (B), where we show the difference between analyses conducted with and without the 1P terms.

7. Page 9 footnote: "This interpretation also reveals that the so-called curled shape of laterally deflected wakes (see e.g. Martínez-Tossas et al. (2021) and references therein) is nothing else than the effect of the horseshoe vortex structure generated behind a lifting wing, albeit with the addition of the swirl caused by the rotor rotation."
This explanation is exactly the one provided by Shapiro et al. (2018) [1] which the authors have not referenced in their study. This statement should be removed and references to Shapiro et al. (2018) [1] must be added.

We have added the requested reference. However, the footnote in reality was simply expanding on a citation of a classical book on helicopter theory (Johnson, 1995). In fact, the idea of seeing a misaligned rotor as a finite span wing is decades old, and classically taught in helicopter courses to scores of students. As this concept predates Shapiro et al. 2018 by many years, a classic as Johnson 1995 seems to be a much better fit in this case.

8. Equation (13): This is the same lateral velocity equation derived by Shapiro et al. (2018) [1] derived using Prandtl's lifting line theory, except that Shapiro et al. (2018) [1] assumed that thrust varies with cos(yaw)^2. Heck et al. (2023) [2] extended the lifting line model of Shapiro et al. (2018) [1] to no longer assume thrust follows cos(yaw)^2, and the final answer in Heck et al. (2023) [2] (Eq. (2.15)) is the same as Eq. (13) here.

We have added a reference to Heck et al. (2023), and we stress here once again the dramatic difference between that work and the present approach: the former lacks a generic thrust model.

9. Equation (14): Similarly, this is the same induction model derived by Heck et al. (2023) [2], although it is presented in a slightly different form in the previously published paper. Appendix E from Heck et al. (2023) [2] is pertinent (i.e. combine Eq. (2.15) with Eq. (E1) to arrive at the induction model form below that can be compared to Eq. (14) in the present paper). From Heck et al. (2023) [2], the induction model equation is:

$$a_n = \frac{2C_T - 4 + \sqrt{16 - 16C_T - C_T^2 \sin^2(\gamma)}}{-4 + \sqrt{16 - 16C_T - C_T^2 \sin^2(\gamma)}}$$

which appears to yield identical predictions to Equation (14) in this study.
In summary, Section 2.5 is a repeat of existing literature and can be removed, with appropriate references added.
We believe that a complete derivation of the relevant equations is necessary here, in order for this article to be understandable and self-contained. The references to the relevant literature are provided in the manuscript, including references and extensive comparisons with Heck et al. 2023.

10. Line 206: Please justify the assumption of the small inflow angle, especially in the context of yaw and tilt misalignment and shear.
The choice of a small inflow angle is used for simplifying the equations. Clearly, this approximation somehow limits the applicability of the model, as stated in the accompanying text. However, we have now added an example that shows this angle to be small for a typical operating condition. Additionally, the extensive results and comparisons with LES and experimental measurements -in a wide variety of operating conditions- further support this choice.

11. Equations (18a) and (18b): Have the authors assumed that the lift and drag coefficients are constant along the wind turbine blade? Please explain.
Not exactly, these model parameters represent "equivalent" values that render the effects of the actual spanwise-variable quantities on the global power and thrust produced by the rotor. This approach is similar to classical ones widely used for helicopter rotors (Jonhson, 1995). We have expanded on this concept in the next Sect. 3.3 and 3.4, and provided extra results in Sect. 4.2.

12. Line 239: "The power model reveals that vertical shear is the culprit for the observed lack of symmetry with respect to yaw misalignment." I don't quite understand this sentence. When the authors state "observed lack of symmetry," are they referring to existing published literature or to their own data (which to this point has not been presented). Previous studies have already explained and modeled that wind speed shear and wind direction veer cause the asymmetric power with respect to yaw misalignment. This current study neglects veer, which also seems limiting.
This sentence seems rather clear to us. We are not aware of other previous studies, including the most recent Heck et al 2023, that can explain why the power of a misaligned rotor is non-symmetric with respect to negative and positive yaw misalignments. On the other hand, our model presents a clear analytical expression of the effect of shear, and we discuss this at length in Sect. 2.8 and in the results Sect. 4.
Any effect of veer is a different topic, and not discussed here. As mentioned in an earlier reply, veer -together with non-standard shear, low level jets, horizontal shear etc.- could be readily

added to the proposed model. However, we consider these as interesting material for a follow-up study.

13. Figure 5: The effect of wind speed shear is very small. I expect the effect of veer is much larger, and especially when there is both shear and veer.
The reviewer's assertion that the effect of veer is much bigger is speculative at best, and would require specific investigations to the proven or rejected. As noted above, we consider this topic to be out of scope for this study, which seems to be ample enough.

14. Paragraph beginning on line 274: It's good the authors state assumptions and simplifications here, but they should also all be stated and explained within the derivation. Otherwise, it seems as if the authors are making ad hoc choices about what is important and what approximations are made.
We are not sure what is meant here: of course we make hypotheses, assumptions and choices in order to derive the model, as in any theory. We also try to explain as clearly as possible all our choices, and we repeat them when we believe it is necessary for clarity. In the lines mentioned here by the reviewer, we have simply repeated the limits of the model. However, we have revised the text, and -as previously noted when replying to other remarks- added various further comments on the assumption of small angles and given reasons for neglecting the tangential induction and other effects.

15. Equation (28) and associated discussion: I do not understand the motivation for simplifications to be applied to the model and then the tuning of more unknown parameters. How does this affect the result? How can this be done in general? Do the authors expect these parameters to be universal, and if not, how can this model be applied to a new wind turbine model? Do we need power data for turbines in yaw misalignment to tune this model? If so, that is not necessarily useful as a predictive model.
We have expanded the discussion on this important point, adding the new Sect. 3.3 and 3.4, where we propose a simplified choice of the model parameters. Furthermore, we have demonstrated the procedure in the new Sect. 4.2, testing the model on the NREL 5 MW and on a modified DTU 10 MW reference turbines. These examples replicate what one would do in the absence of dedicated tuning data, showing the general applicability of the proposed approach and its predictive ability.

16. The authors need to include an Appendix that describes the tuning process in much more detail. What do the authors mean when they say "a different random 50% subset of the available data"? Is this the training-testing split? Presumably the authors are not performing model tuning with the same data that are used to test the model accuracy, as this is improper practice and can bias the results. In the added Appendix I am requesting, model results without any tuning must be shown.
We expanded the text and clarified the tuning approach, which now has a dedicated new Sect. 3.3. Additionally, we have added a new experimental dataset that was purely used for validation purposes and not for tuning.

17. β: It strikes me as a bit strange to have a tunable parameter in the model represent a known geometrical feature such as blade twist.
It is not strange, as the model requires an "equivalent" value of twist, as for other blade characteristics. We have expended the text, providing an explanation in Sect. 3.4 of where twist should be measured (which, as for helicopter rotors, falls at about 2/3 of the blade span). We have also verified this value on four different wind turbines, as now explained in the new Sect. 4.2.

18. Figure 7: the authors show four results in this figure that are almost identical. It is very challenging to discern any notable differences among the subfigures, so it's reasonable to ask whether the authors have really tested the limits and applicability of their modelling framework. For example, why has the tip-speed ratio been kept within such a small region?

Thank you for the comment. The numerical analysis was based on datasets that were available to us at the time. Now good part of the results based on LES comparisons has been completely rewritten, and it includes a new complete set of conditions in region II, III, and in transitions between the two. Additionally, a wider range of tip speed ratios is considered in the experimental part of this work.

19. Line 324: "However, as thrust is decreased (and pitch increased), power capture at positive γ values is larger than for negative misalignments." The asymmetry is almost not visible in Figure 7 to me. Perhaps quantify to make it more clear?

Thank you. We have added the new Figures 10 and 18 to better show the effects of shear.

20. Line 327: "trust coefficient" -> "thrust coefficient"

Thank you. We have corrected the sentence.

21. Figure 8: Why have the authors chosen to only consider low thrust coefficients with a maximum of C_T=0.6? It is interesting to show higher thrust coefficients. For example, Heck et al. (2023) [2] found that the thrust should increase with yaw to reduce the power loss. However, it seems that their induction model (same as your Equation (14)) is less accurate at higher C_T.

The C_T that was reported in fig. 8 was the average one for plus/minus 30 degrees of yaw, hence the relatively low values. The ranges were selected to have a complete dataset. In fact, data for interpolation of higher C_Ts was not available for TSR<9.5. This figure is no more present in the revised manuscript, and replaced by fig. 10.

22. Line 341: "[...] and is capable of describing even relatively minor effects of the complex behavior of a misaligned wind turbine rotor in a sheared inflow." I did not follow what the authors meant by this statement

We mean that -even though the physics of misaligned rotors is quite complex- a simple model like ours is able to correctly capture the trends and phenomena involved. The sentence seems correct and understandable to us.

23. Line 345: Since the rotor aerodynamic characteristics are necessary in your model, please include an Appendix which describes all relevant characteristics in this paper, so that the paper is self-contained.

We have now added Sect. 3.4, which explains how the model parameters can be computed from the aerodynamic data of the turbine. Since we use four different wind turbines, to avoid excessively increasing the length of the article, we decided not to report these data, which can be found in the relevant (cited) bibliographical references and associated online repositories.

24. What is the thrust coefficient of the experimental turbine?

The experimental turbine model operated at a wide range of thrust coefficients, depending on the control strategy and on the controller inputs. The text has been expanded for clarity.

25. Figure 11: It seems that a lot of tunable parameters are fit within this model. I am again unclear as to what is within sample of tuning and what should be considered as model validation (which requires out of sample data).
We have added an additional experimental dataset that was purely used for validation purposes and not for tuning. The new version of the text is hopefully now clearer on these aspects of our work.

26. Figure 13: The results are summarized as having "very good" agreement with limited discussion, but there are several occasions where the model predictions are outside experimental uncertainty. It would be better to discuss these in detail.
The critical points that do not lie within the uncertainty bands are obtained with very low tip-speed ratios. In these conditions the small angle approximation is not accurate, which impacts the accuracy of the results. The text has been expended to explain this fact.

27. Section 4.1: The results of this section align exactly with the published study of Heck et al. (2023) [2], who found that the thrust coefficient should be increased as the yaw is increased to reduce the power loss.
Thank you for this comment. We have now mentioned the agreement with Heck et al. (2023).

28. Section 4.2: How does the operation of the leading turbine affect the wake? The specific equations should be shown. It should affect the initial streamwise and spanwise wake deficits. This was done in Heck et al. (2023) [2] (Figure 9 and Appendix C).
As noted by the reviewer, this was explained in Heck et al 2023. Therefore, we prefer not to repeat the analysis here, as the paper is already quite long.

29. Figure 15: It seems that there is almost no benefit from the modified model compared to baseline FLORIS, from the lines in Figure 15(f). Why is the additional benefit so negligible?
The benefit is on the order of 1% power gain, which, in our opinion is not negligible. The reason for this is the trade-off generated by increasing the C_T upstream. On one hand, more power can be produced upstream, but at the same time the wake expands and presents lower momentum, thereby impacting the production downstream.

30. Line 444: "The LES-ALM results confirm the findings based on the FLORIS engineering wake model: less power losses for the front turbine, and more gains for the downstream one." Looking at Figure 15, this statement appears to be incorrect. It seems that the "Opt. (Model)" approach actually increases the turbine 1 power loss and increases turbine 2 power gain. Rather than decreasing power loss for the front turbine with more gains for the downstream one.
Correct, thank you for pointing this out. We have corrected the text accordingly.

31. References: I am not sure why the authors have chosen to cite arXiv versions of papers that have been published before this present paper was submitted, but that must be corrected.
Thank you for pointing out this. We have edited this accordingly.

32. I recommend a title change, since other published papers have previously gone 'beyond the cosine law.' It is better to be specific about what contributions this paper contains.
We respectfully disagree, the title clearly indicates the content of the paper. The differences with other previous publications are now very extensively described in the revised manuscript, and could not be captured by a title, which clearly has to be relatively short.

[revised manuscript text omitted]

---

## Author Response (AR2)

**Reply to the reviewers' comments for paper "On the power and control of a misaligned rotor – Beyond the cosine law"**

The authors would like to thank the two reviewers for their time and for the useful feedback.

A reply to the comments of Reviewer #2 is reported in the following. The reviewer's comments are in black, and our replies in blue.

A revised version of the manuscript is attached to the present reply, with additions highlighted in blue and deletions marked in red. We have taken this opportunity for making small editorial changes to some captions, and for correcting a wrong symbol.

The authors

Reviewer 2

1. Figure 10: The authors may consider describing the mechanism that causes the asymmetry from shear to reduce at higher thrust coefficients
   Thank you for your suggestion. The mechanism was already explained in Sect. 2.8, but we have now expanded the discussion to make it clearer.

2. Figure 15: The caption and figure seem to refer to different variables, please double check.
   Thank you for spotting this! We have now corrected the caption.

[revised manuscript text omitted]